# Zebrafish pigment cells develop directly from persistent highly multipotent progenitors

Tatiana Subkhankulova[1,12], Karen Camargo Sosa [1,12], Leonid A. Uroshlev [2], Masataka Nikaido[1,9], Noah Shriever [1], Artem S. Kasianov [2,3,4], Xueyan Yang[1,10], Frederico S. L. M. Rodrigues [1], Thomas J. Carney [1,11], Gemma Bavister[1], Hartmut Schwetlick[5], Jonathan H. P. Dawes [5], Andrea Rocco [6,7], Vsevolod J. Makeev [2,3,8] & Robert N. Kelsh [1] ✉

Neural crest cells are highly multipotent stem cells, but it remains unclear how their fate restriction to specific fates occurs. The direct fate restriction model hypothesises that migrating cells maintain full multipotency, whilst progressive fate restriction envisages fully multipotent cells transitioning to partially-restricted intermediates before committing to individual fates. Using zebrafish pigment cell development as a model, we show applying NanoString hybridization single cell transcriptional profiling and RNAscope in situ hybridization that neural crest cells retain broad multipotency throughout migration and even in post-migratory cells in vivo, with no evidence for partially-restricted intermediates. We find that leukocyte tyrosine kinase early expression marks a multipotent stage, with signalling driving iridophore differentiation through repression of fate-specific transcription factors for other fates. We reconcile the direct and progressive fate restriction models by proposing that pigment cell development occurs directly, but dynamically, from a highly multipotent state, consistent with our recently-proposed Cyclical Fate Restriction model.

Neural crest cells (NCCs) are highly multipotent stem cells. A long-standing controversy exists over the mechanism of NCC fate restriction, specifically regarding the presence and potency of intermediate progenitors. The direct fate restriction (DFR) model, based on early in vivo clonal studies, hypothesised that intermediates are absent and that migrating cells maintain full multipotency, before directly differentiating into single fates driven by signals in their post-migratory environment[1–4]. However, most authors favour progressive fate restriction (PFR) models, with fully multipotent early NCCs (ENCCs) transitioning

[1]Department of Life Sciences, University of Bath, Claverton Down, Bath BA2 7AY, UK. [2]Vavilov Institute of General Genetics, Russian Academy of Sciences, Ul. Gubkina 3, Moscow 119991, Russia. [3]Department of Medical and Biological Physics, Moscow Institute of Physics and Technology, 9 Institutskiy per., Dolgoprudny, Moscow Region 141701, Russia. [4]A.A. Kharkevich Institute for Information Transmission Problems (IITP), Russian Academy of Sciences, Bolshoy Karetny per. 19, build.1, Moscow 127051, Russia. [5]Department of Mathematical Sciences, University of Bath, Claverton Down, Bath BA2 7AY, UK. [6]Department of Microbial Sciences, FHMS, University of Surrey, GU2 7XH Guildford, UK. [7]Department of Physics, FEPS, University of Surrey, GU2 7XH Guildford, UK. [8]Laboratory 'Regulatory Genomics', Institute of Fundamental Medicine and Biology, Kazan Federal University, 18 Kremlyovskaya street, Kazan 420008, Russia. [9]Present address: Graduate School of Science, University of Hyogo, Ako-gun, Hyogo Pref. 678-1297, Japan. [10]Present address: The MOE Key Laboratory of Contemporary Anthropology, School of Life Sciences, Fudan University, Shanghai 200438, PR China. [11]Present address: Lee Kong Chian School of Medicine, Experimental Medicine Building, Yunnan Garden Campus, Nanyang Technological University, 59 Nanyang Drive, Yunnan Garden 636921, Singapore. [12]These authors contributed equally: Tatiana Subkhankulova, Karen Camargo Sosa. ✉e-mail: bssrnk@bath.ac.uk

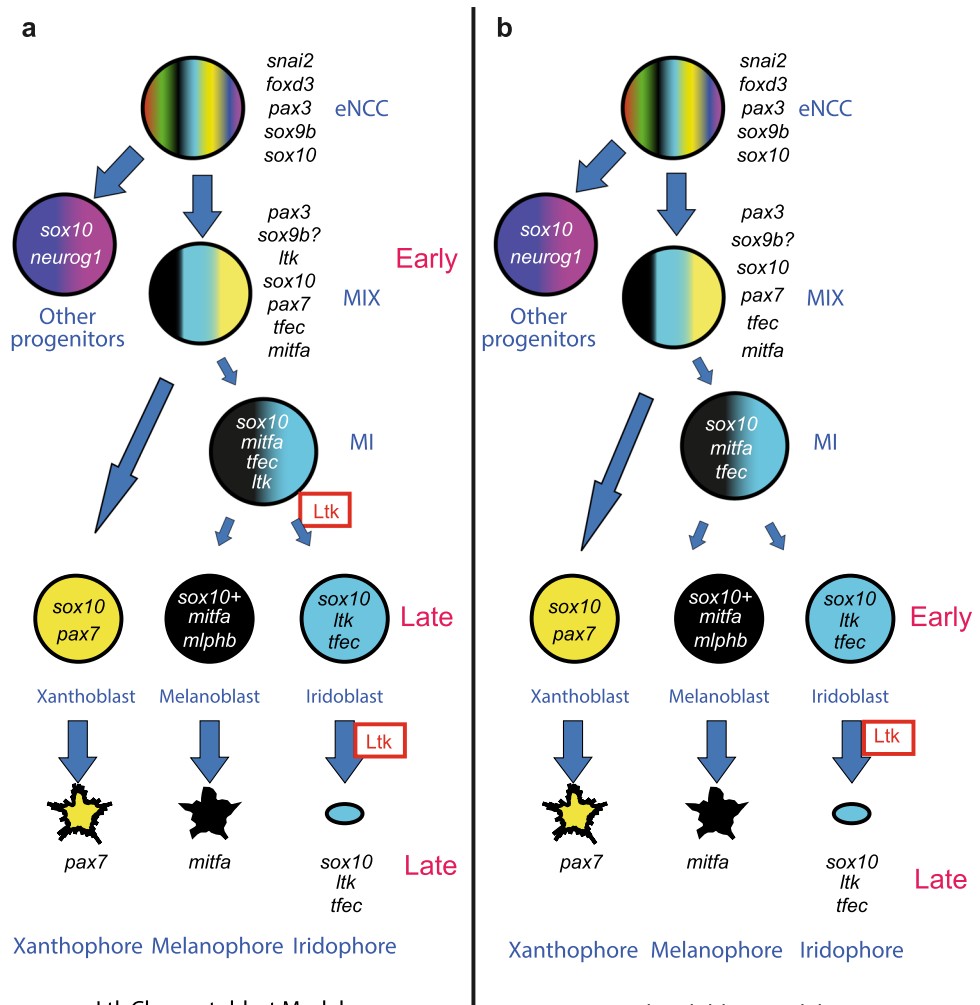

**Fig. 1 | PFR models for zebrafish pigment cell development from neural crest.**
Models show early NCCS (eNCCs) generating multipotent (MIX) and bipotent (MI)
intermediates en route to generating melanocytes and iridophores, and distinguish
timing of expression and role of Ltk signaling: **a** Ltk Chromatoblast Model **b** Ltk
Iridoblast Model. In these schema, potency of cells (number of fill colours) at
different stages in PFR of the pigment cell lineages decreases down the diagram (i.e.
with time), reflecting PFR. Expression of *ltk* is indicated by italics (*ltk*); other key
marker genes are indicated too. Ltk function (signaling activity) is indicated in
Roman script (Ltk, boxed). Thus, in the Ltk Chromatoblast Model, *ltk* is expressed
in all chromatoblasts (MIX) and melanoiridoblasts (MI)('Early'), but Ltk signaling is

activated only in a subset of them, driving iridophore lineage specification (irido-
blast specification). Continuing *ltk* expression in iridoblasts and iridophores ('Late')
has *late* role in iridophore differentiation, proliferation and/or survival. In the Ltk
Iridoblast Model, early phase expression represents iridoblasts ('Early'), where it
functions in differentiation or survival. Late expression reflects ongoing expression
and ongoing function in differentiated iridophores. Experimental assessment using
Ltk inhibitor treatment reveals early role in iridophore fate specification and later
role in differentiated cells, supporting Ltk Chromatoblast Model (**a**) (Supplemen-
tary Fig. 9).

to partially-restricted intermediates before committing to indivi-
dual fates[5–9].

NCCs generate a diversity of cell-types, including cranial skeleto-
genic fates, peripheral neurons and glia and, except in mammals,
multiple pigment cells[10–13]. Fish pigment cells include black melano-
cytes (M), yellow xanthophores (X), and reflective iridophores (I)[14]. In
zebrafish, pigment cell (*M*elanocyte, *I*ridophore, *X*anthophore) devel-
opment is an important test case, since a PFR model with distinct
multipotent (chromatoblast, MIX) and bipotent (melanoiridoblast, MI)
intermediates has been suggested[15–20]. A long-standing, yet untested,
hypothesis proposes that all vertebrate pigment cell-types share a
common, and exclusive (i.e. able to generate only the three pigment
cell-types, but *not* neural or other fates), progenitor, the chromato-
blast (MIX)[15]. Analysis of mutants for the *microphthalmia-related
transcription factor a (mitfa)* gene, encoding a master regulator for
melanocytes[21], was consistent with a bipotential MI progenitor in
zebrafish[17]. These and complementary studies in medaka[22–24] lead to a

widely-accepted, but untested, PFR model of pigment cell develop-
ment (Fig. 1). In mouse and chick, shared progenitors for melanocytes
and glia have been proposed[25,26], and whilst in mouse migrating NC
cells retain multipotency (defined there as at least bipotency)[27], neural
derivatives appear to originate via a PFR mechanism involving bipotent
neuroglial progenitors[28]. However, distinguishing cells that are
*restricted* to a subset of fates from those *specified* to those fates is non-
trivial in vivo[18,19,29,30]. Transcriptional profiling of cells identifies cell
*specification* state, defined as being where cells show expression of one
or more characteristic markers for that fate, but until recently, this was
judged by one or two markers at a time, leaving unclear whether the
same cells also showed specification towards other fates too. The
advent of single cell transcriptional profiling enabled a comprehensive
assessment of key genes (especially fate determining transcription
factors and receptors for fate specification signals), and hence a more
complete assessment of likely potency, although now sensitivity to low
level expression becomes the key limiting factor.

In this work, single cell transcriptional profiling of zebrafish pigment cell development by NanoString hybridisation identifies only broadly multipotent intermediate states between ENCCs and differentiated melanocytes and iridophores. This observation, combined with optimised RNAscope in situ hybridization, provides evidence for retained multipotency even in post-migratory NCCs in vivo. Leukocyte tyrosine kinase (Ltk) marks the multipotent progenitor and iridophores, consistent with biphasic *ltk* expression, which was previously hypothesized to identify both the chromatoblast and committed iridophore cells[18–20]. Ltk inhibitor and constitutive activation studies support expression at an early multipotent stage, where prolonged Ltk signalling drives iridophore differentiation at the expense of other fates through a mechanism of repression of fate-specific transcription factors. However, lineage-tracing of *ltk*-expressing cells reveals their multipotency extends beyond pigment cell-types to neural fates. We conclude that pigment cell development does not involve a conventional PFR mechanism, but instead occurs directly and more dynamically from a broadly multipotent intermediate state, reconciling the DFR and PFR models and consistent with our recently-proposed Cyclical Fate Restriction (CFR) model[29,30]. We propose that single cell RNA-sequencing (scRNA-seq) studies nicely document these dynamic changes in fate specification and differentiation, but often underestimate fate potential of these cells.

## Results

### NanoString hybridisation single cell profiling of fish NCCs

We investigated the transcriptional profiles of 1317 zebrafish NCCs throughout pigment cell development (18–72 h post-fertilisation (hpf)). To obtain sensitive quantitation of key mRNA expression, we used the NanoString platform and a set of 45 genes, focused on those known or suspected to have important roles in melanocyte or iridophore fate specification and commitment, but including multiple markers of early NCCs, plus key markers of other major NC-derived fates (Supplementary Dataset 1). We profiled cells isolated by FACS (Methods; Supplementary Fig. 1 and 2) from *Tg(Sox10:Cre)^{ba74}xTg(hsp70l:loxP-dsRed-loxP–Lyn-Egfp^{tud107Tg})* embryos at 8 time-points in which NCCs and their derivatives are labelled with eGFP after a brief heat shock (Fig. 2a). Our sample consisted of WT cells isolated at time points between 18–72 hpf (444 experimental, denoted "regular", reg; all numbers are for the cells which survived quality control tests and normalization; see Supplementary Fig. 3), plus four distinct sets of other cells: NCCs isolated by FACS from dissected tails of WT embryos at 24 hpf (expected under default PFR model to be enriched for early NCCs and MIX; 108 WT tail cells); NCCs from *sox10* mutants at 30 hpf (enriched for cells trapped in MIX state[18,20,31]; 132 sox10 mutant cells); and gradient centrifugation-purified melanocytes (19 contM cells) and iridophores (25 contI cells) from 72 hpf WT embryos as controls for the differentiated state of these two cell-types. To identify cells with similar expression profiles we conducted cell clustering by gene expression profiles (see Methods). We employed control pigment cells to obtain a natural measure of gene expression variation within a cell type and selected parameters of the clustering algorithm so that the "control clusters" containing most of the control melanocytes and iridophores respectively, had the largest proportion of these control cells in the respective cluster, arriving at 7 clusters (Fig. 2b, c).

Identification of specific clusters 1–3 was facilitated by a combination of enrichment for control cell-types (Supplementary Fig. 2 and 3) and expression of key markers (Supplementary Fig. 4–7), and these are identified accordingly as three pigment cell clusters (melanocytes, iridophores and xanthophores). Interestingly, the melanocyte cluster contained two clearly distinguished subclusters, both of which included approximately 50% of control melanocytes (Supplementary Fig. 8).

Due to the developmental gradient along the anteroposterior axis, NCCs isolated from tails at 24 hpf are enriched for developmentally early cell-types. One cluster was highly enriched with tail cells (Supplementary Fig. 4a), and showed high level expression of two known early NCC markers (*snail1b, sox9b*)[32–34], but also *tfap2a* and *tfap2e*, both required early in melanocyte development[35](Fig. 2b; Supplementary Fig. 4d, 6 and 7). Consequently, we interpreted this cluster as the earliest stage isolated by our *sox10*-dependent labelling technique (note that *sox10* expression begins after *sox9b* and *snail1b*, which are both downregulated in most *sox10*-expressing cells[20]), naming them early highly multipotent progenitors (eHMP).

We used *slingshot*[36] to build pseudotime trajectories of pigment cell differentiation, starting from the eHMP (Fig. 2c; Supplementary Fig. 5). Trajectories for both melanocyte and iridophore differentiation went through another cluster which also contained some tail cells and showed a similar expression profile; we named these cells late highly multipotent progenitors (ltHMP). The ltHMP cells were distinguished from eHMPs by increased expression of *ltk* and lower levels of *snai2* and *sox9b* (Fig. 2b; Supplementary Fig. 4d, 6 and 7), consistent with the known progression of marker expression in zebrafish NCCs[18–20], and also by a striking decrease of *tfap2e*. In contrast, both eHMP and ltHMP show prominent expression of key fate specification genes (*mitfa, pax7b, tfec, phox2bb, sox10*) for diverse pigment cell and neural fates. Both eHMP and ltHMP cells are identified from all stages, even up to 72 hpf, suggesting that they are not just found in premigratory NCCs, but persist at later stages.

Amongst those cells included within the pseudotime ordering of melanocyte and iridophore development are a few belonging to cluster 7 (Supplementary Fig. 9a), which we were unable to assign to a particular cell type with the selection of marker genes at hand. Supplementary Fig. 4d shows that cluster 7 displays *id2a* expression, which is common in all NCC derivatives, as well as that of some other genes visible at early stages of NCC differentiation (*alx4b, ednrba, her9, hmx4, impdh1b, mc1r, pax7b* and *tfap2a*). This cluster does not express any specific markers of pigment cells, such as *mlphb* and *oca2* (melanocyte), *pnp4a* (principally iridophores, but at lower levels in melanocytes and xanthophores) or *pax7a* (xanthophore). With the set of tested markers we are unable to assign the cells of cluster 7 to any of the other cell types, although they presumably include various fate-biased cell-types distinguished by marker genes not included in our gene set. Cluster 4 lacks most genes in the NanoString gene-set, except *foxp4, her9, id2a*, and *impdh1b*, with some expression of the early *tfap2a* gene, and is most likely much more heterogeneous than is apparent with our marker set with its limited representation of neural and skeletogenic fate markers.

Interestingly, when *sox10* mutant cells are included, we observed that they were not separated from WT cells, but occupy part of the central cloud (Supplementary Fig. 4b) and are effectively excluded from the differentiated pigment cell clusters. As expected given the strong failure of pigment cell fate specification in *sox10* mutants[11,18–20,31], mutant cells generally lack pigment cell markers (e.g. *pnp4a, pax7a, mlphb, oca2, pmel, slc24a5*, and *tyrp1b*) and show somewhat reduced expression of some fate specification (e.g. *mitfa*) and early markers (e.g. *sox9b*). They show retained expression of other early markers (e.g. *snai2*) and broadly-expressed genes (e.g. *id2a, impdh1b, ednrba, alx4b*)(Fig. 2e; Supplementary Fig. 4d). In general, *sox10* mutant cells are more homogeneous than WT cells and form a single cluster if optimal parameters of cluster validation for WT clustering are used. Broadly, their expression profile resembles the HMP clusters seen in WTs. Fifty-two percent (70/135) of *sox10* mutant cells show detectable *ltk* with 35% (47/135) showing elevated levels of *ltk* (Fig. 2e), comparable to the ltHMP cluster (Fig. 2b), and in agreement with semi-quantitative observations by whole-mount in situ hybridization of a subset of cells trapped in a premigratory position, with other neural progenitors migrating as normal on the medial pathway[11,20]. Together, these observations are consistent with the previous deduction that mutant cells with the potential to adopt

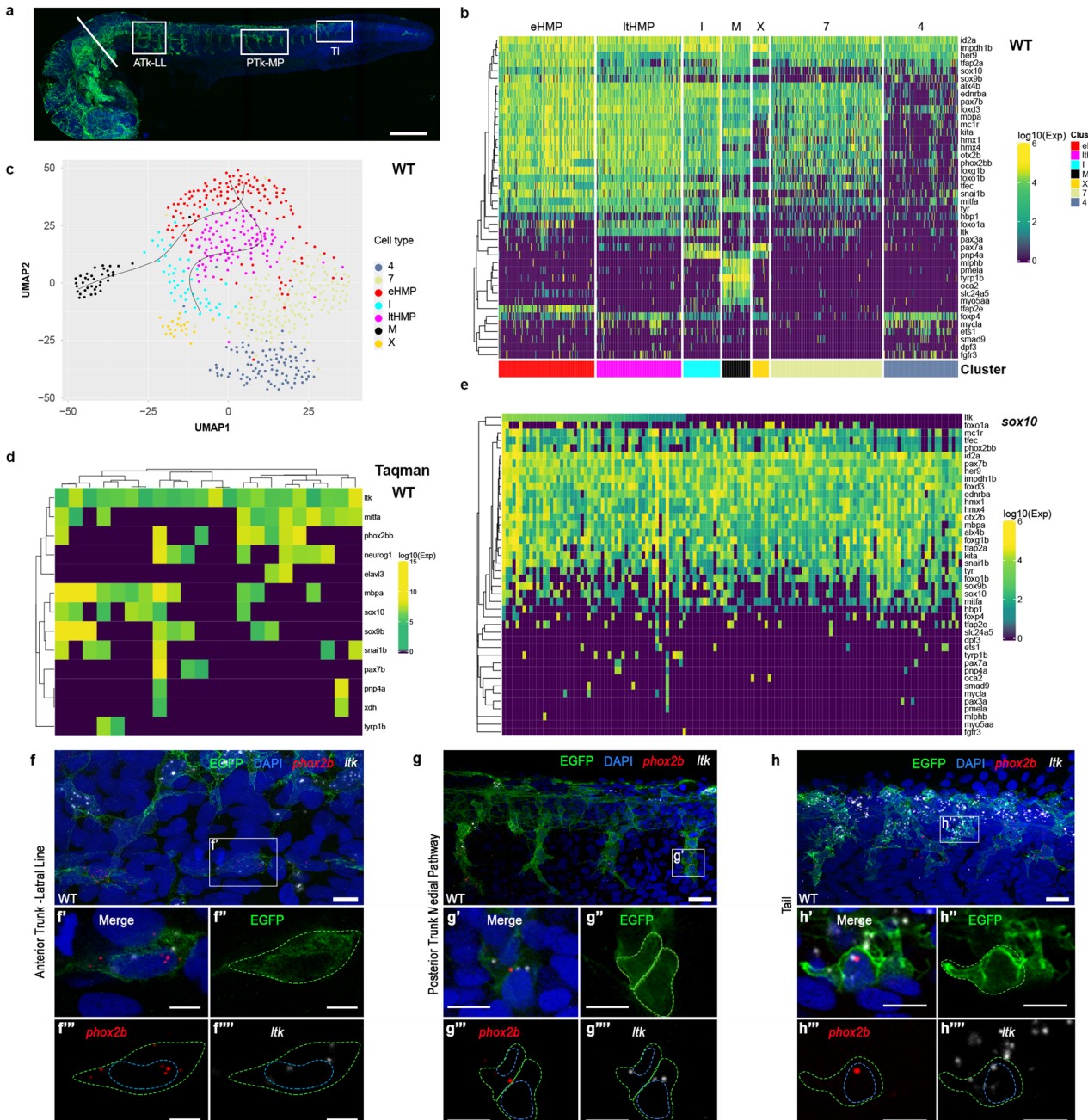

**Fig. 2 | Highly multipotent, but not partially fate-restricted, intermediates are readily detected in zebrafish embryos. a** Whole-mount immunodetection of EGFP in *Tg(Sox10:Cre)^ba74^xTg(hsp70l:loxP-dsRed-loxP −Lyn-Egfp^tud107Tg)* showing source of single cells for NanoString profiling (24 hpf stage shown here; embryos at 30+ hpf had heads removed by severing behind otic vesicle (white bar). White boxes imaged in close-up in panels **f**–**h** to show anterior trunk lateral migration pathway and posterior lateral line nerve (ATk-LL), posterior trunk medial pathway (PTk-MP) and tail (Tl). **b**–**e** Single cell profiling of zebrafish NCCs reveals unexpected absence of partially-restricted pigment cell progenitors. **b** Heat maps showing NanoString profiles of clusters identified in **c**; see also Supplementary Fig. 3 for violin-plot representation. **c** NCC profile clustering, clustered and visualized in 2D by UMAP. Clusters are identified by distinctive expression profiles: eHMP (red), early highly multipotent progenitors; ltHMP (magenta), late highly multipotent progenitors; I (cyan), iridophores; X (gold), xanthophores; M (black), melanocytes; clusters marked with numbers not identified due to lack of distinctive marker gene profiles. Pseudotime trajectories to melanocyte and iridophore are indicated (see Supplementary Fig. 5) **d** Heat map showing independent qRT-PCR (Taqman assay)

assessment of overlapping fate specification gene expression in 24 hpf WT embryos. Only cells with detectable *ltk* expression shown; for full profiles of all cells, see Supplementary Fig. 6. **e** Heat map of *sox10* mutant cell NanoString transcriptional profiles, ordered by *ltk* expression levels. Due to low levels of ltk expression, cells with at least 5 gene counts are shown, instead of 30 in other panels. **f**–**h**, RNAscope expression analysis reveals co-expression of pigment cell and neuronal fate specification genes in vivo in 27 hpf *Tg(Sox10:Cre)^ba74^xTg(hsp70l:loxP-dsRed-loxP−Lyn-Egfp^tud107Tg)* embryos after brief heat-shock to express EGFP. **f, g, h**, RNAscope FISH detection of *phox2bb* (red) and *ltk* (white) is shown in confocal lateral view projections of the lateral line (LL) in the anterior trunk (ATk; **f**), medial pathway (MP) in the posterior trunk (PTk; **g**) and tail (Tl; **h**). Insets (f'-h') show co-expression of *phox2bb* and *ltk* in EGFP-labelled cells. Merge and individual channels for a single focal plane for each inset is shown in f'-h'''. Yellow dashed line shows membrane cell border. Blue dashed line shows outline of nucleus for NCCs as revealed by DAPI (blue in merge). Scale bar dimensions: a =200 μm, g and h =20 μm, f, g'-g''' and h'-h'''=10 μm, and f'-f''' =5 μm.

pigment cell fates (indicated in vivo by expression of *tfec*, *ltk*, *pax7b*, and *tyr*; Fig. 2e; Supplementary Fig. 4d) are prevented from proceeding along the pigment cell differentiation pathways, and remain in a multipotent progenitor state characterised by expression of *ltk*, *tfec* and *sox10*[18–20,31]. However, the NanoString data reveals that most cells of even the *ltk*+state also express *phox2bb* (Fig. 2e; Supplementary Fig. 4d), indicating a previously unanticipated potential to adopt at least some neuronal fates as well, although this potential is not realized in vivo because it is blocked by the absence of functional Sox10[37]. Interestingly the expression profile of *sox10* mutant cells is also close to that of cluster 7, except that expression of various genes, including *snai1b* and *sox10*, is elevated in *sox10* mutant cells, again perhaps indicating that they are trapped in an early progenitor state of some form. Further exploration of the precise state of the *sox10* mutant cells will require examination of an expanded series of markers.

As an independent validation of the overlap of *ltk* and other pigment fate specification genes with early neuronal fate specification genes, we used TaqMan qRT-PCR to evaluate a further sample of more than 100 single NCCs, isolated by FACS from 30 hpf WT embryos, examining both *phox2bb* (sympathetic and enteric neuron fate specification) and *neurog1* (sensory neuron) expression (Fig. 2d; Supplementary Fig. 10). Out of 21 cells expressing detectable *ltk* (with Ct≤29), 12 (57%) showed detectable *mitfa*, whilst 12 (57%) showed one or both of *phox2bb* and *neurog1*; if we include the Schwann cell marker *mbpa* then 17 (81%) of *ltk*-expressing cells show one or more of these neural fate markers (Fig. 2d). Examining the full data-set, we note how 79% and 85% of cells expressing *phox2bb* or *neurog1* respectively express at least one of the key pigment cell fate specification genes *ltk, mitfa* or *pax7b* (Supplementary Fig. 10). Together our data suggests the unexpected possibility that broad multipotency is retained in most progenitor cells.

## Comparison of NanoString profiling and published scRNA-seq

The disadvantage of NanoString profiling technology is that it depends on the limited number of preprepared probes and thus may display accidental clustering resulting from random matching of a small number of marker genes. To rule out this possibility we used data obtained in a series of recent genome wide scRNA-seq studies in zebrafish, many of which focused on different aspects of pigment cell development. In scRNA-seq data the distance between cell expression profiles is defined by all genes detected in the transcriptome, rather than from a limited number of probes, so that cells accidentally adjacent in the NanoString hybridization profiles would be expected to be distant from each other in the scRNA-seq transcriptome profiling.

We downloaded read counts of seven scRNA-seq profiling studies[38–44](see Supplementary Table 1), conducted batch correction and normalization and plotted cell distributions as UMAP plots (Supplementary Fig. 11). Most of the cell cloud in the UMAP plot contained mixtures of cells coming from different studies, yet we identified two areas populated with the cells from a single study: Wagner[41], which used ß-actin2 as a marker and thus contained many cell types other than NCC, and Saunders[44] which included later stages of development. Inspection of the UMAP areas populated with cells expressing specific known markers (Supplementary Fig. 12) readily identifies regions populated with pigment cells (all three types), Schwann cells, and neurons (Supplementary Fig. 13). We tested distributions of all genes included in our NanoString hybridization study and observed clear co-expression of identical cell type markers in the scRNA-seq study (*alx4a, id2a, impdh1b, her9, pnp4a, sox10*, and at low levels *foxd3, ltk* and *tfec* in iridophores; *her9, mitfa, pmela, pnp4a, slc24a5, sox10, tyrp1b* and at low levels *kita, tfap2a* and *tyr* in melanocytes; *her9, impdh1b, mitfa, pax7b, pnp4a, sox10*, and at low levels *myo5aa, pax7a, tfap2e in* xanthophores; *alx4a, foxd3, foxo1a, foxp4, her9, sox10*, and at low levels *id2a, impdh1b, mitfa, mycla, tfap2a* in early NCCs). In scRNA-seq the cells with the active transcription of these genes were found

near the cells expressing other known gene markers for particular cell types, not included in our NanoString probe sets (e.g. *alx4b* for iridophores, *dct* for melanocytes, *csf1ra* for xanthophores, see Supplementary Fig. 13). Thus, we concluded that despite a limited number of markers used to make NanoString probes the resulting distribution of cell types agreed well with the distribution of the cell-types obtained from scRNA-Seq profiling, at least for pigment and eNCC cells, the markers of which were primarily included within the set of NanoString probes.

The only gene which displayed a strong disagreement in NanoString and scRNA-Seq profiles was tfap2e, which appeared as a clear eNCC marker in the NanoString profile, but in the scRNA-seq profiling reflected melanocyte-specific expression, in agreement with a number of previous studies.

Interestingly, many key fate specification and determination genes were poorly represented in our integrated scRNA-seq studies (e.g. *ltk* and *tfec* (iridophores), *phox2bb* (enteric and sympathetic neural cells), *kita* (melanocytes)(Supplementary Fig. 12). Some genes (e.g. *id2a, mitfa, pax2b*) displayed a much broader expression profile in NanoString hybridization experiment than in RNA-seq profiling. We conclude that NanoString hybridization was more sensitive for weakly-expressed genes, but probably displayed saturation effects for more highly-expressed ones (so that relative levels of expression are less accurately detected in those cell types in which transcription was active at relatively high levels).

We conclude that whilst these studies document changing transcriptional profiles associated with differentiation in great detail, likely reflecting the environmental signals the cells encounter, they are less well-equipped to assess the retained potency of these cells. Taken together, these data and our NanoString profiles suggest a more dynamic situation than strictly envisaged in the DFR and PFR models, where NCCs retain high multipotency, whilst becoming biased towards specific fates by environmental signals; we have recently proposed a Cyclical Fate Restriction (CFR) hypothesis to reconcile these observations[29,30].

## Co-expression of key fate specification genes in situ

To further validate these unexpected findings, we used our optimised RNAscope in situ hybridization protocol to examine co-expression of pigment cell and neuronal fate specification genes. We focused on testing *phox2bb* co-expression patterns that were unexpected under the PFR model. Expression of *phox2bb* in zebrafish has been previously reported as restricted to NCCs migrating along the developing gut from c 24 hpf onwards, interpreted as progenitors of the enteric ganglia, and sympathetic ganglial progenitors in the ventral medial pathway from 36 hpf[45,46]. Analysis of *phox2bb* morphants suggests it is required for fate specification of enteric and sympathetic neurons[37,46]. Using RNAscope, we detect *phox2bb* expression in 27 hpf zebrafish within many premigratory NCCs, but also some migrating NCCs on the medial pathway, and even in many NCCs associated with the posterior lateral line nerve (Fig. 2f–h). Furthermore, in all of these locations, cells also expressing *ltk*, a key iridophore specification gene, are readily found[20](Fig. 2f–h; Supplementary Fig. 14). Thus, *phox2bb* expression can be detected at low levels in cells from premigratory stages, long before formation of sympathetic or enteric ganglial progenitors, but consistent with the expected autonomic neuron potential of premigratory NCCs (Fig. 2h; Supplementary Fig. 14i). More strikingly, that autonomic neuron potential is then maintained in a widespread manner, including both ventralmost cells on the medial migration pathway (Fig. 2g; Supplementary Fig. 14g, h;[47]) and in putative Schwann cell progenitors associated with the lateral line nerve (Fig. 2f; Supplementary Fig. 14f). The expression in putative Schwann Cell Precursors of the posterior lateral line nerve is intriguing, suggesting retained autonomic neuron potential that is unlikely to be realized in vivo. We extended our study to later time-points, detecting expression of *ltk* or

*phox2bb* in cells of the posterior lateral line nerve at both 36 and 50 hpf (Supplementary Fig. 15b, d). Co-expression of *ltk* and *phox2bb* remained detectable even up to 50 hpf in some cells in premigratory positions and migrating on the medial pathway, but was not detected at 3 dpf (Supplementary Fig. 15a, c). At 50 hpf *phox2bb* was detected in putative sympathetic and enteric neurons, but also (and unexpectedly) in cells associated with the DRG and spinal nerves. Quantitative analysis of the co-expression of *ltk* and *phox2bb* revealed that numbers of co-expressed transcripts tended to be decreased at later stages (Supplementary Fig. 15a).

### Expression of *ltk* in multipotent progenitors and iridoblasts

Expression of *leukocyte tyrosine kinase* (*ltk*) was particularly striking in our single cell profiles, showing relatively high expression in both ltHMPs and iridophores (Fig. 2b). Previous studies have shown expression of *ltk* in a subset of WT NCCs at premigratory stages, with maintenance and upregulation in specified iridoblasts and differentiating iridophores[19,20]. Taking advantage of our optimized RNAscope protocol, we confirm and expand these observations (Supplementary Figs. 16 and 17). The biphasic pattern of expression, initially in subsets of premigratory and migrating NCCs, and becoming restricted to cells of the iridophore lineage from around 2 dpf was confirmed (Supplementary Fig. 16). Furthermore, quantitation of *ltk* transcripts in tail NCCs of WT and *sox10* mutants at 24 hpf revealed that *sox10* mutants showed a statistically significant increase in *ltk* expression levels over WTs of same age, although absolute levels in individual premigratory cells were highly variable (Supplementary Fig. 17e). Furthermore, our quantitation at 2 and 3 dpf showed that WT iridophores showed a significantly elevated level of *ltk* expression compared to premigratory NCCs at 24 hpf, consistent with previous suggestion of upregulation of ltk in differentiating iridophores (Supplementary Fig. 17bv,e–i);[20]. In the context of previous work, this suggested a series of further tests of the model emerging from our single cell expression profiling, as now described.

*ltk* encodes a receptor tyrosine kinase, and analysis of loss- and gain-of-function mutations indicated that Ltk signalling drives iridophore fate specification[20,48,49]. We had postulated two phases of expression of *ltk* in NCCs: Late phase expression (from around 26 h post-fertilization (hpf) in the posterior trunk, later in tail) corresponds to the differentiating iridophore lineage, while early phase expression (around 22–24 hpf in the trunk) in premigratory NCCs represents multipotent progenitors (Fig. 1a)[18–20]. An alternative model postulates that both early and late phase *ltk* expression is in cells committed to the iridophore lineage (Fig. 1b). Interestingly, in our NanoString profiles *ltk* shows covariance with two clusters of genes (iridophore markers (e.g. *pnp4a, impdh1b*) and broadly expressed genes; Supplementary Fig. 18), suggesting biphasic expression, consistent with this proposal. To test directly the biphasic expression model, we used a small molecule inhibitor of Ltk[48] to disrupt Ltk signalling in zebrafish embryos at different stages; allowing for the spatiotemporal differences in NCC development along the anteroposterior axis, we find distinct effects of inhibition of early and late phase expression, with the former restricting the number of iridophores specified, and the latter controlling the expansion of these cells, likely by proliferation to form clones (Fig. 3a–m). As expected, homozygotes for a putative null *ltk* allele, *ltk^ba7*, show essentially no iridophores (similar to findings for the *ltk^ty82* allele before[50]) and this is not changed when treated with Ltk inhibitor throughout the 18-68 pf window (Fig. 3j–m).

### Ltk signalling drives other cell fates to iridophores

As a further test, we predicted that, if Ltk activity drives iridophore fate from multipotent progenitors, expression of constitutively active Ltk signalling in NCCs would promote ectopic and supernumerary iridophore formation at the expense of other fates. We used constitutive activation of Ltk through generation of an N-terminal fusion with

human Nucleophosmin, NPM-Ltk (Fig. 4a), which drives NCCs to an iridophore fate when expressed using the zebrafish *sox10* promoter[48]. We generated a plasmid, *Tg(Sox10:NPM-ltk, egfp)* (hereafter, *Tg(sox10:NPM-ltk)*) which uses the *sox10* promoter to drive expression of both NPM-Ltk and the lineage tracer, enhanced GFP (Fig. 4a). As a negative control, we created a dead kinase (DK) version of *sox10:NPM-ltk* by exchanging lysine 943 in the kinase domain of wild type zebrafish Ltk protein with arginine (K943R, Fig. 4a)(plasmid *Tg(Sox10:NPM-ltk_K943R, egfp)*, here referred to as *Tg(sox10:NPM-ltk(DK))*) based on the previously reported activity in human *ALK*[51]. To assay the fate of constitutively expressing cells, we focused on whether the *Tg(sox10:NPM-ltk)* construct might redirect cells from a melanocyte to an iridophore fate, because the melanocyte fate is readily characterized by colour, and because the reciprocal fate switch is hypothesised to underlie the increased number of iridophores documented in *mitfa/nacre* mutants[16]. Whilst *Tg(sox10:NPM-ltk)* readily induced precocious (Fig. 4c) and ectopic (Fig. 4c',c") iridophores at this stage, these were almost undetected at 60 hpf in embryos injected with *Tg(sox10:NPM-ltk(DK))* (Figs. 4b, 4b'), showing that *Tg(sox10:NPM-ltk(DK))* has minimal NPM-Ltk activity (Supplementary Table 2). We injected embryos with *Tg(sox10:NPM-ltk)* or with *Tg(sox10:NPM-ltk(DK))* and scored embryos having normally elongated body axis at 60 hpf for the presence or absence of additional iridophores, and for GFP-positive melanocytes (Supplementary Table 2; Fig. 4d–f'). As expected, in control embryos injected with DK version, we did not find embryos with additional iridophores. In the 50 *Tg(sox10:NPM-ltk(DK))* injected embryos that were GFP-positive, we identified 11 GFP-positive melanocytes (Table 1; Fig. 4e, e'). In striking contrast, we found only one GFP-positive melanocyte in the 53 GFP-positive embryos injected with *Tg(sox10:NPM-ltk)*, a significant reduction (Table 1; Chi-squared test, $P = 0.00233$). We conclude that active Ltk signalling in NCCs is largely incompatible with melanocyte development, instead driving cells to adopt an iridophore fate. Consistent with this interpretation, the single melanocyte obtained in an embryo injected with *Tg(sox10:NPM-ltk)* was poorly differentiated, being small, round and less dendritic than those in control embryos, and was found in an embryo that had no additional iridophores, suggesting weaker activation of Ltk signal therein. These data are consistent with the idea that Ltk signalling in a multipotent progenitor cell drives iridophore fate choice at the expense of other fates. To further probe the mechanism, we analysed injected embryos at 28 hpf, quantitating the frequency with which cells expressing the *ltk* constructs expressed key transcription factors driving specific fates (Supplementary Fig. 19). Strikingly, cells expressing the activated NPM-Ltk construct show a dramatic increase in the proportion of cells expressing *tfec*, and correspondingly decreased *mitfa* expression compared to controls expressing the NPM-Ltk(DK) construct (Supplementary Fig. 19a, b). However, equally striking decreases in the proportion of cells expressing *pax7a* and *phox2bb* were also observed in response to activated Ltk signalling (Supplementary Fig. 19a, b). Together, these data show that Ltk signalling is sufficient to drive NCCs to adopt an iridophore fate at the expense of other pigment cells and autonomic neurons, and that this happens through a mechanism of repression of key fate-specific transcription factors for these fates.

### *ltk*-expressing cells generate non-ectomesenchymal NC fates

To test directly the proposed multipotency of *ltk*-expressing cells, we used fate-mapping using transient expression of an *ltk:gfp* transgene, examining the prediction that green fluorescent protein (GFP) expression under the *ltk* transcriptional regulatory regions should label all pigment cells, but also other (e.g. neural) derivatives as well. We used BAC recombineering techniques[52,53] to make a reporter construct for the *ltk* gene, *TgBAC(ltk:gfp)* (Fig. 5a). To confirm that the *ltk:gfp* transgene faithfully reproduces the endogenous *ltk* expression pattern, we combined immunodetection of GFP with RNAscope

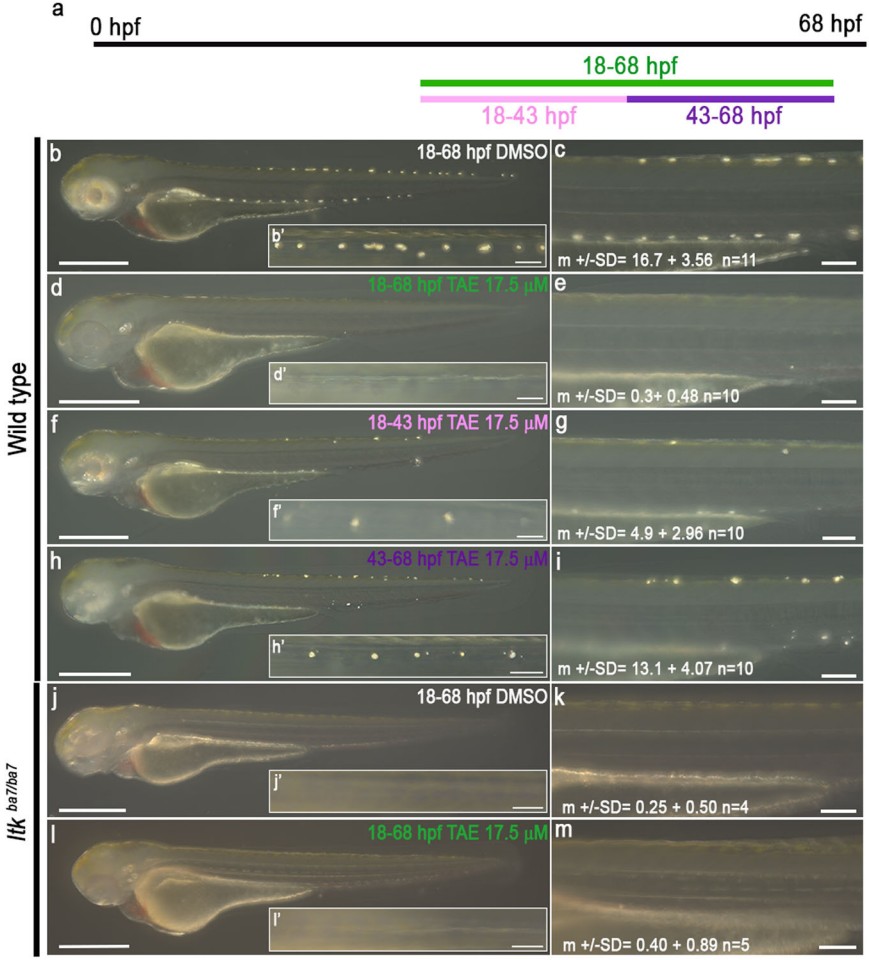

**Fig. 3 | Inhibition of Ltk activity during early and late treatments results in 2 distinct phenotypes.** Schematic showing time window treatment (**a**). Embryos were treated with DMSO (**b** and **c**) or 17.5 μM TAE-684 (**d**–**i**) during defined time periods: 18-68 hpf ('control', b-c), 18-68 hpf ('full', **d**-**e**), 18-43 hpf ('early', **f**-**g**) and 43-68hpf ('late', **h**-**i**). Iridophores in the dorsal stripe were detected through their endogenous reflectivity; PTU-treatment from 24 hpf was used to suppress melanisation. Note how 'early' inhibition of Ltk reduces the number of iridophores (specification defect), whilst 'late' inhibition prevents the enlargement of iridophores (proliferation or differentiation). Homozygous *ltk* mutants (*ltk*[ba7/ba7]) treated with DMSO (**j** and **k**) or 17.5 μM TAE-684 (**l**-**m**) during 18-68 hpf were indistinguishable, showing no more than occasional iridophores in the dorsal stripes, and similar to WT embryos treated in the same time window (**d** and **e**). All images were taken left side view with dorsal to the top. **b**–**h** n = 10 embryos on each condition; n = 3 independent experiments. **j** n = 4 embryos, **l** n = 5 embryos; n = 3 independent experiments. Source data are provided as a Source Data file.

detection of endogenous *ltk*; GFP expression is seen to overlap with endogenous *ltk* expression in premigratory NCCs (Supplementary Fig. 20a-d[iv]) and in cells with a spatial distribution consistent with them being differentiating iridophores (Supplementary Fig. 20e-f[iii]). As a positive control construct to test we could label each NCC derivative when GFP was expressed in early NCCs, we used a *sox10:gfp* PAC (P1 phage-derived artificial chromosome; see Methods for details)(Fig. 5a). We injected each of these two constructs into *Tg(neurogenin1(8.3):rfp)* transgenic fish, in which red fluorescent protein (RFP) is expressed in developing dorsal root ganglion (DRG) neurons in embryos from as early as 2 dpf[54] to facilitate identification of GFP reporter expression in early stage DRG neurons, since our previous studies showed that *sox10*-driven GFP perdurance in DRG does not usually last beyond c. 48 hpf[55]. The presence of GFP expression in all NCC-derived cartilage, pigment cell and neural cell-types was assessed (see Methods for details of criteria). The *TgPAC(sox10:gfp)* construct labelled all expected neural crest derivatives as found by single cell labelling studies (Table 2; Fig. 5c)[12,13,31]. The results for embryos injected with the *TgBAC(ltk:gfp)* construct were consistent with our hypothesis (Table 2; Fig. 5b). Whilst many GFP-positive iridophores were seen in embryos injected with the *TgBAC(ltk:gfp)* construct, GFP expression was clearly

not restricted to this cell-type, consistent with our expectation that early phase *ltk* expression represented a multipotent progenitor. Strikingly, we saw expression of GFP in numerous other pigment cells, including both melanocytes and xanthophores, but also in glial and neuronal derivatives, specifically DRG sensory neurons and enteric neurons. This observation indicated other NCC-derivatives, such as glial and neuronal cells, are also derived from early *ltk*-positive cells.

## Discussion

How NCCs adopt individual fates from their extensive repertoire has remained an enigma. Recent adoption of a PFR model has been most well-worked out for the adoption of pigment cell fates in zebrafish, where two distinct key intermediates were proposed (Fig. 1). Our data provides direct assessment of aspects of this model, with the unexpected conclusion that the identified progenitors (eHMP and ltHMP) both show a transcriptional profile including all key fate specification genes assessed, strongly indicating broader multipotency than expected under the PFR model; conversely, both expected partially-restricted intermediates (MIX and MI) were not detectable in vivo. We and others have shown the crucial role for Ltk signalling in fate specification of iridophores[20,48,49]. Here we demonstrate experimentally

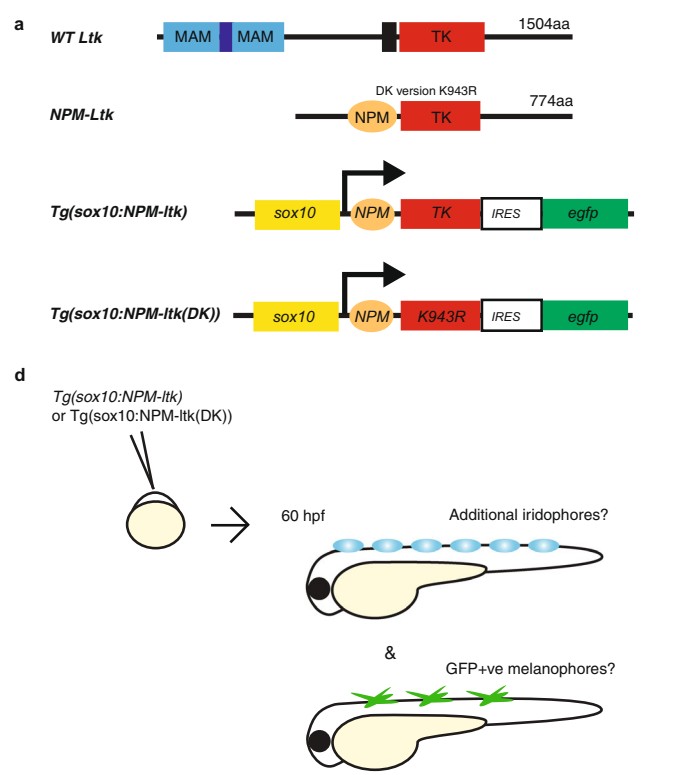

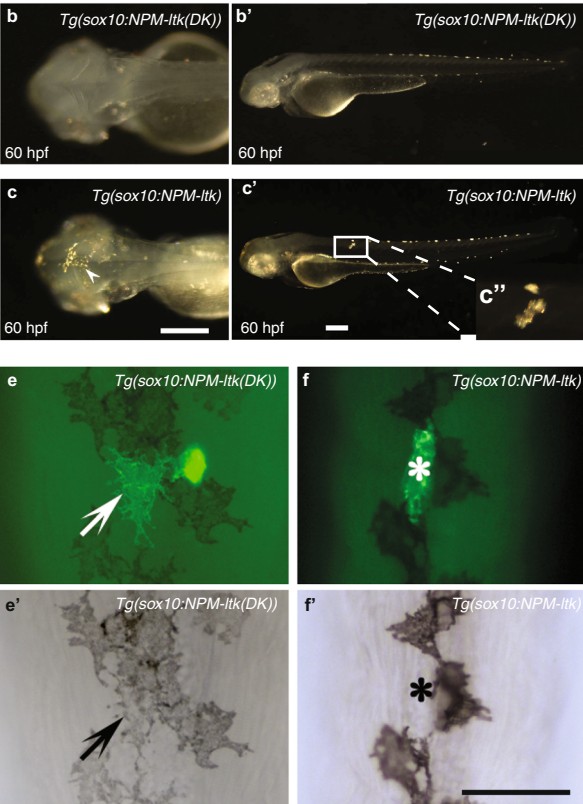

**Fig. 4 | Active Ltk signaling is incompatible with melanophore development.**
**a**, Schematic drawings of wild-type Ltk and NPM-Ltk fusion protein, and of NC expression construct *Tg(sox10:NPM-ltk)*; see text for details of these and of negative control kinase dead construct, *Tg(sox10:NPM-ltk(DK))*. **b-c'**, Validation of constitutively-activated Ltk and dead kinase control. Embryos injected with *sox10:NPM-ltk DK* (**b, b'**) and *Tg(sox10:NPM-ltk)* (**c, c'**), imaged at 60 hpf. Precocious (arrowhead in **c**) and ectopic (inset **c''**) iridophores are shown. Melanisation was inhibited by PTU treatment to enhance detection of iridophores. **b, c** Dorsal views with anterior to the left. **b', c'** left side views with dorsal to the top. Scale bars: 100 μm. **d-f'**, Ltk activity is inconsistent with melanocyte differentiation.

**d**, Schematic drawing of experimental procedure. DNA constructs were injected into embryos at 1-cell stage. Embryos were cultured until 60 hpf, scored for precocious/ectopic iridophore formation, and subjected to anti-GFP antibody staining. **e-f'**, GFP-positive melanophore in control embryos injected with *Tg(sox10:NPM-ltk(DK))* (arrows in **e, e'**); a second GFP-positive cell was an iridophore based upon its position and shape. In embryos injected with *Tg(sox10:NPM-ltk)*, GFP-positive cells were almost invariably not melanised (asterisks in **f, f'**). All views show dorsal midline, anterior to the top. Panels **e** and **f** are fluorescent images merged with bright field images and **e'** and **f'** are bright field images for **e** and **f**, respectively. Scale bar: 50 μm.

## Table 1 | Activated Ltk signalling suppresses melanophore formation

| | No. of observed embryos | No. of embryos showing GFP-positive cells | No. of GFP-positive melanophores in GFP-positive embryos[a] |
|---|---|---|---|
| *Tg(sox10:NPM-ltk(DK))* | 215 (100%) | 50 (23%) | 10 |
| *Tg(sox10:NPM-ltk)* (+irido)[b] | 95 (100%) | 37 (39%) | 0 |
| *Tg(sox10:NPM-ltk)* (-irido)[(2)] | 142 (100%) | 16 (11%) | 1 |

Notes: [a] Embryos were raised in a low dose of PTU (25% of normal dose, i.e. 0.00075%) to partially inhibit melanin synthesis, so that GFP fluorescence in the living cells of injected embryos could be readily assessed in melanophores.
[b] "+/-irido" here means presence or absence of precocious/ectopic iridophores in injected embryos, respectively.

that early phase expression of *ltk* in premigratory NCCs is necessary for this fate choice decision, and that Ltk signalling can suppress other fates, through a mechanism of transcriptional repression of key transcription factors (e.g. *phox2bb, pax7*) determining alternative fates. Lineage-tracing of the fate of these early phase *ltk* expressing cells demonstrates that early phase *ltk* expression does indeed mark broadly multipotent progenitors, which generate glia and all types of pigment cells, but also at least some neuronal cells. An alternative explanation, that the *ltk* expressing NCCs are a mixture of cells of distinct potencies (i.e. broad multipotency is true only of the population, but not of individual cells) is inconsistent with our single cell profiling data, which readily identified a large cluster of cells showing

overlapping expression of key genes underpinning diverse fate choices, including neural fates, reflecting their broad multipotency. A corollary of this observation is the insight that fate specification of an individual cell-type depends as much on repression of inappropriate fate-specific transcription factors (e.g. *phox2bb* in a melanocyte), as on upregulation of appropriate ones (e.g. *mitfa*)[18,19,30]. This view is supported by the observation from a recent scRNA-seq study of zebrafish neural crest using an *mitfa:Cre* driver line to label cells expressing *mitfa*, which identified clusters corresponding to all pigment cell, neuronal and glial cell-types[43].

We have been unable to identify clusters that would correspond to either the MIX or MI intermediates previously hypothesized. An

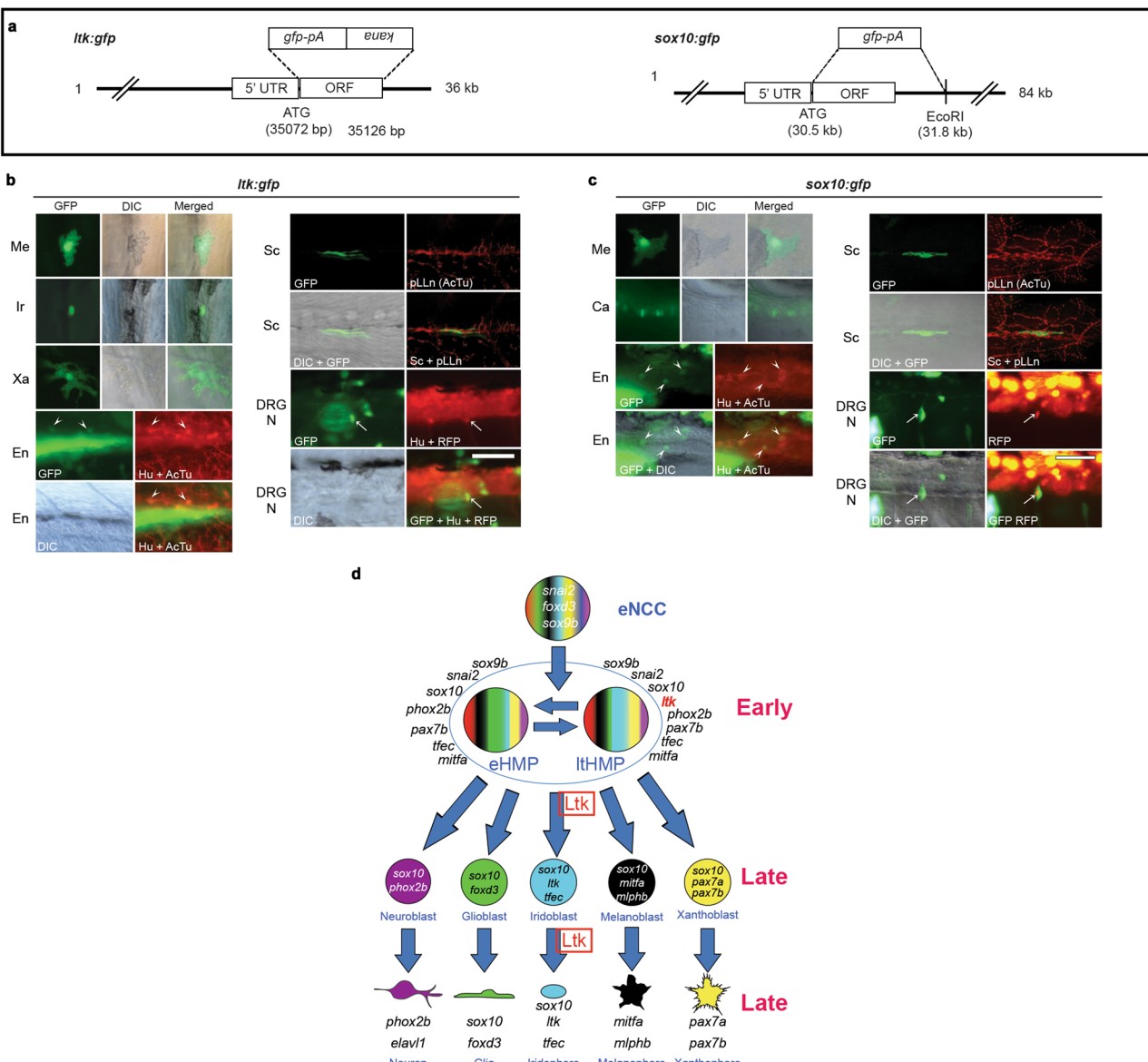

**Fig. 5 | Genetic fate mapping using a *TgBAC(ltk:gfp)* construct identifies pigment cells, but also Schwann cells and neurons, as derived from *ltk*-expressing cells. a** Schematic drawings of reporters. In both constructs, *gfp* cDNA with polyA site (*pA*) is inserted at the site of the first methionine. See Methods for details. **b** Cells labeled by *TgBAC(ltk:gfp)*, shown at 2 dpf (DRG neuron), 3 dpf (melanocyte) or 4 dpf (others). All pigment cells are labeled by GFP or anti-GFP antibody, and observed by differential interference contrast (DIC) and immunofluorescence (GFP) microscopy; combined DIC and fluorescent images also shown (merged). Schwann cell around the posterior lateral line nerve is also detected by anti-GFP antibody (GFP), and pLLn is detected by anti-acetylated tubulin antibody (AcTu). Enteric and DRG sensory neurons are labelled by GFP, with enteric neurons identified by position (revealed by Differential Interference Contrast (DIC)) and anti-Elavl1 staining (Hu)(arrowheads; anti-AcTu also detected), whilst DRGs are

identified by position and anti-Elavl1 staining (Hu) and RFP driven by *neurog1* promoter (arrows) (see Methods). DRG N, DRG neuron; En, enteric neuron; Ir, Iridophore; Me, melanocyte; pLLn, posterior lateral line nerve; Sc, Schwann cell; Xa, Xanthophore. **c** Cells labeled by *TgPAC(sox10:gfp)*. In addition to pigment cells and Schwann cells, lower jaw cartilage (Ca), enteric neurons and DRG are also labeled. For quantitation, see Table 2. Scale bar: 50 μm for **b** and **c**. **d** Cyclical Fate Restriction model of pigment cell development. Pigment cells derive from Highly Multipotent Progenitor (HMP) cells, which are envisaged as cycling through multiple sub-states; for simplicity, only two of these (eHMP and ltHMP) are distinguished here, based principally upon the level of expression of *ltk* (see Fig. 2b). Early and late phase Ltk expression reflects that in the ltHMP and iridoblast/iridophore respectively, with Ltk function (Ltk, boxed) required early for specification of iridophore lineage from ltHMP and late for ongoing differentiation/proliferation. See Fig. 1 legend for key.

alternative explanation is that our limited gene set lacked the key markers, although this seems unlikely since the markers used were mostly selected for their known crucial roles in fate specification and determination of neural crest-derived cell-types and we show here that Ltk signalling at early stages is sufficient to drive iridophore fate development through a mechanism of repression of key transcriptional regulators of alternative fates. Furthermore, our analysis of

published scRNA-seq studies demonstrated that cell profiles identified from thousands of transcriptionally active genes formed very similar groups duly expressing known iridophore, melanocyte and xanthophore markers. Interestingly many markers with key contributions for cell-type identification in our NanoString profiling (*ltk* and *tfec* in iridophores, *oca2* and *tyr* in melanocytes, *pax7a* in xanthophores) displayed low level, sporadic and poorly localised expression in

**Table 2 | *ltk*-expressing cells generate neural and all pigment cell-types**

|  | Melanocyte | Xanthophore | Iridophore | Glia | DRG neuron | Sympathetic neuron | Enteric neuron | Cartilage | Total NCC |
|---|---|---|---|---|---|---|---|---|---|
| *TgPAC(sox10:gfp)* | 222 (24%) | 625 (68%) | 8 (1%) | 27 (3%) | 22 (2%) | 0 (0%) | 14 (2%) | 7 (1%) | 925 (100%) |
| *TgBAC(ltk:gfp)* | 7 (4%) | 86 (47%) | 64 (35%) | 10 (5%) | 9 (5%) | 0 (0%) | 7 (4%) | 0 (0%) | 183 (100%) |

Notes: All figures are cell number, and figures in parenthesis are percentage out of total number of NCC-derivatives. We counted only the cells that have RFP or Hu as neurons. Some other cells appeared likely to also be neural crest-derived neurons, such as enteric or sympathetic neurons, because of their position, but these were not included here unless marker was positive. However, neuron marker negative cells in DRG position were counted as "DRG glia" because they were considered to be either undifferentiated neural precursors or satellite glia, and are included in Glia (19 for *sox10* and 3 for *ltk* constructs respectively). In case of *ltk* (but not *sox10*) reporter we saw frequent ectopic labeling of Rohon-Beard neurons (total=146), posterior lateral line neurons (181) and trigeminal neurons (160). Abbreviations are the same as those used in Fig. 3.

scRNA-seq profiling. We believe that scRNA-seq protocols may miss genes transcribed at low levels, which yet can be identified in Nano-String profiling. Our observations agree well with those published by Soldatov et al. in mouse[28], where they were unable to identify many key transcription factors in their scRNA-seq profiles, but could infer their presence from coordinated changes of transcription activity of groups of genes activated by these transcription regulators. Of course, this is necessarily restricted to expression at levels that is functionally significant. On the other hand, for low level gene expression, NanoString profiling agrees better with low throughput studies. Detection of these genes, even in the absence of their dramatically impacting the cell transcriptionally, is particularly important when defining likely cell potencies at early stages in the specification/differentiation process. These scRNA-seq studies[38–44] show clearly the dynamic change of transcriptional profiles, identifying cells showing co-expression of markers of two or more distinct fates, presumably reflecting the impact of local environment in *specifying* cells to appropriate fates; we argue that our data clearly indicates that these cells have broader *potency* than is readily apparent in these profiles. Indeed, in one of these studies[39], use of sensitive in situ hybridization techniques has shown other examples of cellular colocalisation of unexpected combinations of markers (e.g. skeletal and melanocyte), albeit in relatively few cells, complementing the examples we have generated using RNAscope, and providing further support for the retention of broader potency for which we argue here. In the absence of evidence for the pigment cell-specific intermediates proposed before, we conclude that pigment cell fate specification proceeds directly from a broadly multipotent progenitor.

We note that these HMP progenitors were identified at all stages examined, even at later stages (42-72 hpf) when NCC induction is thought to have long ceased. We hypothesise therefore that these HMP cells include presumed satellite glia and Schwann cell progenitors (SCPs) readily detected using *sox10* expression in association with both ganglia and peripheral nerves[55,56]. We note that at least one population of these, associated with the DRG, has been identified as a population of adult pigment stem cells that generate the post-metamorphic pigment pattern of adult zebrafish[57]. Using an optimized RNAscope protocol we have demonstrated the unexpectedly early and widespread expression of *phox2bb* in premigratory and migrating NCCs, including in a well-studied population of putative SCPs. We note that enteric neurons have been deduced as originating from trunk (likely spinal nerve) SCPs[58], consistent with our interpretation of our observations. It is likely our HMP progenitors correspond in mammals to the SCPs that form the source of many adult melanocytes;[25] indeed, SCPs have also been shown to be a source of parasympathetic neurons and adrenomedullary cells and hence have retained a broad multipotency[59–61].

We reconcile our new view of NCCs retaining high multipotency with the more conventional observations interpreted under a PFR model in former studies (e.g[17–19,42,44].) by noting technical differences in our approaches. Firstly, our use of NanoString technology and a refined RNAscope protocol has given increased sensitivity to low level gene expression compared with scRNA-seq and conventional WISH. For example, expression of *ltk* in the progenitor cells in this study is rather variable, reflecting the notably low levels of WISH expression in the 'early' phases[18–20], and consistent with the apparent absence of *ltk* detection in recent scRNA-seq studies[42,44]. Furthermore, the meta-analysis of zebrafish scRNA-seq studies focused on pigment cell development that we performed here clarifies that many of the key marker genes that we used to deduce potency are poorly-detected in the scRNA-seq studies, but demonstrate clear cell-type specific profiles, consistent with that expected from genetic studies, confirming the enhanced sensitivity of the single cell NanoString and TaqMan RNA-seq assays employed here. This limitation of scRNA-seq is true even in the key mouse studies, despite their higher sensitivity (>7000 genes detected per cell)[28,62]. Our recent expanded WISH characterization identified co-expression with *ltk* of *sox10*, *tfec* and *mitfa* as characteristic of the putative chromatoblast (MIX)[18,19], and we see these genes as more robust components of the ltHMP progenitor identified by our single cell transcriptional profiling here. Comparison of the WT and *sox10* mutant *ltk* expression patterns (here and[20]), together with the highly variable levels of *ltk* in premigratory cells (Supplementary Fig. 2e), strongly argues for highly dynamic changes in *ltk* expression at these stages, which then becomes stabilized as cells differentiate as iridophores, and which, in *sox10* mutants, become stabilized at a higher expression level in cells that do not migrate medially (neural progenitors). In our *ltk* fate mapping studies reported here the prolonged perdurance of GFP expression compensates for the low level and transient expression of *ltk* itself, allowing sensitive detection of the fate of *ltk*-expressing cells. Secondly, differences in single cell isolation protocols used may have acted alongside this elevated sensitivity. We note that our profiles for control pigment cells include unexpected expression of fate specification genes for other fates (e.g. *phox2bb*). In our studies of co-expression of *phox2bb* and *ltk*, although spatially widespread, only a subset of cells showed overlapping expression in vivo even with our optimized RNAscope protocol. Together these observations suggest that the transcriptional profile of our single cells has 'relaxed' away from the apparently specified state seen in vivo by WISH, towards a more basal progenitor state during isolation. This provides a strong indication of the highly dynamic nature of the fate specification and differentiation process – that apparently specified and even differentiating cells identified in vivo are held in states biased towards one or more fates by environmental signals. It also provides evidence that these cells, even when strongly differentiated, were not yet committed at these stages. Thus, our protocol was apparently sensitized to detection of cell *potential*; in contrast, many former studies favoured detection of the state of fate *specification* and progressing differentiation of these cells. Such studies of development of the peripheral nervous system show a pattern of co-expression of competing fate-biases, which are subsequently resolved during differentiation[28,62]. Finally, the ready detection of a broad suite of genes (including quantitative measurements of transcriptional activity for those expressed at higher levels), makes scRNA-seq excellent for observing the transcriptional changes associated with the gradual differentiation process from these multiply-specified cells, as they first stabilize (dominant expression corresponding to one specific fate) and lockdown (commitment) their fate-specific expression profiles.

Our data reinforce certain key observations in recent studies of mouse neural crest development, including the dynamic changes in transcriptional state, influenced by environmental factors, and phases of co-expression of markers for multiple fates en route to differentiation[28,62]. In the latter case the authors emphasise the pairwise nature of these combinations, whereas in our study (and, to some extent, in the zebrafish scRNA-seq studies[38–44]) we are seeing evidence of these co-expressions extending to >2 cell-types, but this discrepancy may reflect the different sensitivities to low level gene expression, and current programmes used for single cell pseudotime ordering. The mouse studies imply commitment to a single fate may occur earlier, but our data suggests this is not the case, because isolated differentiated melanocyte and iridophore control cells show detectable expression of markers for neural fates. However, one prominent feature of our study, the significant expression of *ltk* in premigratory and migrating NCCs is clearly not conserved in mouse[62]. As we show, this expression reflects its role in the specification and commitment of iridophores; this cell-type is not found in mammals, so the absence of expression in the neural crest is consistent with, and may even causally explain, this evolutionary change in NCC potency.

To reconcile our observations with those underpinning the PFR model, we propose a Cyclical Fate Restriction (CFR) model in which our broadly multipotent intermediate is transcriptionally dynamic, transitioning between sub-states, each of which is likely biased towards a subset of derivative fates, but also under substantial control by environmental factors, so that the dwell-time in each biased substate may be extended or minimized appropriately (Fig. 4d; described in detail in[29,30,63]). We propose that heterogeneity of HMP cells results from the intrinsically dynamic nature of their gene regulatory network (GRN), but also the influence of those environmental signals. This would also explain the substantial heterogeneity of marker expression in premigratory and migratory NCCs that led to the PFR model (e.g[64–66].). These dynamic changes might be complex or irregular, but we favour the idea that they show a cyclical progression between states biased towards individual fates. This would provide one explanation for some key observations underpinning the PFR Model of zebrafish pigment cell development: why mutations that prevent the adoption of one specific fate (e.g. melanocyte (*mitfa*)) result in an increase in another specific fate (iridophore), since this would reflect the order in which GRN sub-states biased towards different fates are dynamically organized in the cycle[30]. It is becoming clear that a more dynamic view of stem cells may be required to understand their biology and we believe this will prove true for neural crest stem cells too; sensitive direct or indirect readouts of GRN state transitions will be required to test this model in the future. We note that two of the progenitor clusters identified in our profiling (eHMP and ltHMP) show noticeably similar profiles, and speculate that these might represent two such sub-states; this idea is consistent with the main differences between the eHMP (low *ltk*; high *tfap2a, tfap2e, sox9b*) and ltHMP (high *ltk*; lower *tfap2a, sox9b*; low *tfap2e*) cell clusters identified here, but will require comprehensive experimental testing. Our data shows that constitutively-activated Ltk signalling drives iridophore differentiation, so that an Ltk+ HMP is receptive to environmental signals driving iridophore fate[67]. However, our data here (NanoString profiling, RNAscope quantitation of *ltk* expression) reinforces our earlier deduction[20] that *ltk* expression is variable and highly dynamic. Hence, we predict that HMPs are only transiently sensitive to these environmental signals, but also that the levels and persistence of *ltk* expression may be altered by the spatial environment of an HMP, affecting their bias towards adoption of iridophore fate.

HMP cells appear to persist into later embryonic stages; we speculate that these cells therefore include the persistent neural crest stem cells deduced from studies of pigment cell regeneration and adult pigment cell development[57,68,69], but also that they include many other NCCs which in vivo, depending upon their local environment,

occupy a variety of states of fate specification, with overlapping markers for two or more cell-types[18,19], giving the impression of PFR. Interestingly, a detailed scRNA-seq study of mouse neural crest development showed broadly multipotent progenitor cells, with subsequent neural intermediates interpreted as bipotent progenitors[28]. However, the same study appeared to show markers of melanocyte potential (e.g. *Mitf*) and a Schwann cell marker (*Mbp*) widely distributed amongst many cells, suggesting a pattern consistent with the view we are proposing here. Indeed, a subsequent study from the same group[62] presents a picture of a 'hub' of Schwann cell precursors as a source of numerous NCC-types, with these originating from fate-biased hub cells; importantly these cells are already in distinct anatomical locations, and hence will be experiencing distinct environmental signals. We propose that these hub cells are likely the equivalent of some of our HMPs, and speculate that their environmental influences generates consistent biases in their profile. An interesting test would be to remove them from these environments and place them elsewhere - dynamic movement between these biased sub-states would be predicted under our CFR model. But we note that our HMPs likely include cells progressing down the individual differentiation pathways, but which have not yet reached commitment. Taken together, our investigations of Ltk activity, NCC fate-mapping and single cell profiling show that zebrafish pigment cell progenitors show unexpected and robust multipotency, suggesting a new paradigm for neural crest development encompassing a dynamical view of multipotency that may resolve the long-standing controversies in this area.

Like the view from studies of the mouse peripheral nervous system[28,62], our data suggests a very dynamic process. However, in contrast to that view we are suggesting that retained full multipotency persists even into differentiation stages, rather than commitment being programmed earlier. Such a view may be widely applicable in the context of other progenitor cell-types. For example, analysis of neural plate border cell expression profiles reveals an analogous situation of co-expression of markers for each of epidermis, neural tube, placode and neural crest[70]. In the case of embryonic stem cells, dynamic switching between states has been proposed[71–73], not unlike the dynamic HMP state we are proposing. In our theoretical work we found that it is surprisingly easy to generate CFR dynamics from conceptual GRN models motivated by mutual inhibition[73]. Hence from a theoretical systems biology viewpoint, a dynamical view of fate specification is perhaps not as complex or unorthodox as it might at first glance appear. A key future test of our model will be to look for cyclical changes in the expression of a subset of very early fate-specification markers, including *ltk*, either in vivo or in cells cultured in a neutral environment.

## Methods

### Ethics statement
This study was performed with the approval of the University of Bath ethics committee and in full accordance with the Animals (Scientific Procedures) Act 1986, under Home Office Project Licenses 30/2415, 30/2937 and P87C67227.

### Fish husbandry
Embryos were obtained from natural crosses and staged according to Kimmel et al[74].

All fish were housed according to FELASA recommendations[75], in tanks filled with circulating system water at $28 \pm 0.2\,°C$. System water was made up from Reverse Osmosis water, with synthetic sea salt added at an amount such that the conductivity of the water was kept at approximately 800 uS/cm. Average water quality data were as follows: pH: 7.30, general Hardness: 160 mg/l CaCO3, Ammonia: 0 mg/l NH3, Nitrite: 0 mg/l NO2, Nitrate: <10 mg/l NO3- and Conductivity: around 800 μS/cm. Light cycle was 14 h light/10 h dark (Lights on at 08:00, off at 22:00 daily). All fish were in general good health and no specific diseases were observed throughout the colony. The fish were fed

Paramecium from 5–15 dpf, followed by Zeigler Larval AP100 powder (MBK Installations, Cat. No. LD150) from day 16–22 dpf. From 23 dpf onwards they were fed Sparos Zebrafeed 200–400 (Sparos) and ZM Brine Shrimp Artemia Cysts Premium 260 grade (ZM Systems). Fish euthanized by schedule 1 killing were given an overdose of an anaesthetic (5 drops of 0.48% MS-222 added to a petridish of water).

## Construction of nucleophosmin-*ltk* fusion gene
Primers to amplify cDNAs corresponding to *NUCLEOPHOSMIN* (*NPM*) domain in human *p80* fusion gene[51,76] and to cytoplasmic kinase domain of zebrafish *ltk* gene[20] were as follows.

XbaI/EcoRI *NPM* upper: 5′-GCTCTAGAATTCATGGAAGATTCGAT GGACATG

*NPM* lower SpeI: 5′-GACTAGTAAGTGCTGTCCACTAATATG
SpeI *ltk* upper: 5′-GACTAGTCTATTATCGTAAGAAGAACCACCTG
*ltk* lower XbaI: 5′-GCTCTAGAGCTACACAGGGGTGACACTCAG
*p80* cDNA for template of *NPM* was kindly provided by Dr Richard Jäger (Uniklinikum Bonn, Germany), and *ltk* cDNA is the one reported by our group[20]. Amplified *NPM* and *ltk* fragments were ligated using SpeI site, and subcloned using XbaI site into modified pCS2+ vector, which contains *sox10-4725* enhancer[77] instead of cytomegalovirus (CMV) promoter, IRES sequence and membrane tethered form of *egfp* cDNA, to form *Tg(Sox10:NPM-ltk, egfp)*. As a negative control, we introduced a point mutation to exchange the conserved lysine with arginine[51] at 212 in ATP binding domain of the chimeric NPM-Ltk protein, to generate *Tg(Sox10:NPM-ltk_K943R, egfp)*.

## Combined whole mount fluorescent in situ hybridization and immunofluorescence
Anesthetised embryos were fixed in 4% paraformaldehyde (PFA, Alfa Aesar, Cat.No. J61984.AK) overnight at 4 °C. PFA was removed and 100% methanol was directly added. Samples were stored at −20 °C until processed. We used the RNAscope Multiplex Fluorescent kit V2 (Bio-techne, Cat No. 323110) following the manufacturer's protocol with some modifications. Methanol was removed from samples, which were air-dried for 30 min at RT. 50 µl of Proteinase Plus was added for 15 min (24 hpf), 20 min (30 hpf), 40 min (50 hpf) and 45 min (76 hpf) at RT and washed with 0.01% PBS-Tween (Sigma-Aldrich, Cat. No. P-1379) for 5 min x 3. Samples were incubated overnight with diluted probes (1:100; *mitfa* (Cat No. 444651), *ltk* (Cat No. 444641), *tfec* (Cat No. 444701-C2), *phox2bb* (Cat No. 505091-C2), *neurogenin1* (Cat No. 505081-C2), *pax7a* (Cat No. 1050611-C4) and *sox10* (Cat No. 444691-C4)). Probes were recovered and samples were washed in 0.2X SSCT for 15 min x 3. We followed the manufacturer instructions for AMP 1-3 and HRP C1-C4 using 2 drops of each solution, 100 µl of Opal 570 (Akoya, Cat. No. FP1488001KT) or 650 (Akoya, Cat. No. FP1496001KT) (1:2500) and 4 drops of HRP blocker. Washing in between these solutions was performed twice at RT for 10 min with 0.2X SSCT prewarmed at 40 °C. Samples were incubated in primary antibody rabbit a-GFP (Invitrogen Cat. No. A11122) diluted (1:750) in blocking solution (0.1% PBTween, Normal goat serum (Vector, Cat. No. S-1000) 5% and 1% DMSO (VWR, Cat. No. 87154),1:750) overnight at 4 °C then washed 3x with 0.1% PBTween for 1 h with agitation. Samples were incubated in secondary antibody Goat a-Rabbit Alexa Fluor488 (Invitrogen, Cat. No. A32731TR) diluted (1:1000) and DAPI (2 mg/ml) diluted 1:100 in blocking solution for 2.5 h at RT and then washed 6 × 30 min with 0.1% PBTween. Samples were stored in 0.1% PBTween at 4 °C until mounting. Samples were mounted in 50% glycerol (Fisher Scientific, Cat. No. BP220-1)/PBS in glass bottom petri dishes (WPI, Cat. No. FD35PDL).

## DNA injection
Purified DNA construct was diluted to 50 ng/µl with water, and injected into cell body at 1-cell stage. Developing embryos were sorted and cultured at 28.5 °C until the appropriate stage. To prepare embryos for in situ hybridization, we added phenylthiourea (Sigma-Aldrich, Cat.

No. P-7629) at 0.003%, whereas a quarter dose was used to allow them to be only weakly pigmented when we needed to detect fluorescent signals in melanocytes after antibody staining.

## Construction of *TgBAC(ltk:gfp)* and *TgPAC(sox10:gfp)* reporter
We identified a fully sequenced BAC clone containing long 5′ flanking region of *ltk* gene (CH211-254C11, 36100 bp, Accession number is CR387922) in Zebrafish Information Network database, ZFIN (http://zfin.org/). The exon-intron structure of *ltk* gene in this BAC was determined by comparing our reported cDNA of *ltk* (accession number EU399812) with this BAC clone. This analysis showed that the BAC contained the coding sequence of the first exon in the area between 35072 bp and 35126 bp, though this coding sequence is longer than the one in the first exon reported in our previous paper[20]. To insert the *gfp* cDNA into this BAC, we used BAC recombineering[52]. The targeting vector, containing *egfp* cDNA with bovine growth hormone polyA and *kanamycin resistant* gene as a selection marker[53], and bacteria (SW101) used for recombination were obtained from Dr. Higashijima and from Biological Resources Branch of Frederick National Library for Cancer Research, respectively. We prepared fragment to be inserted into BAC by high fidelity PCR using targeting vector as template. The primers were as follows.

*ltk*−5arm:
5′-AGAGATTAGGCTAACAAACACTTTATCTCCGGGATCCTTTTTA AGGAGCC-
atggtgagcaagggcgagga
*ltk*−3arm:
5′-TTAACGTTAACAGAAACCAGCAGGCCAGTATTAATTAGCAAA ACACTCAC-
cagttggtgattttgaactt
The sequences in lower case of *ltk*−5arm and *ltk*−3arm target the *gfp* cDNA and *kanamycin resistant gene*, respectively. The sequences in upper case correspond to the *ltk* genomic sequence, and act as homologous domains during recombination. The recombineered resultant was confirmed by PCR targeting the two BAC ends and two junctions between *ltk* gene and *egfp* or *kanamycin resistance* gene. To make transient transgenic embryos, we simply injected 100 pg complete construct into fertilized eggs.

As a positive control construct to demonstrate how readily each NCC derivative was labelled when GFP was expressed in early NCCs, we recombined gfp into an 84 kb P1 phage-derived artificial chromosome (PAC) containing the zebrafish sox10 gene and 30 kb upstream and 52 kb downstream sequences (clone BUSMP706I16137Q2). To create this TgPAC(sox10:gfp) construct, we used RecA-dependant recombineering[78] as described in[79]. Characterization of TgPAC (sox10:gfp) transient transgenics confirmed that it accurately reproduced the early sox10 expression pattern, labelling all premigratory NCCs[79], and thus behaved similarly to a plasmid construct containing 4.9 kb of upstream sequence characterized previously[55,77].

## Generation of new *ltk* mutant allele
WT fertilized eggs were injected with 2 nl of RNP complex targeting exon 2 of the *ltk* gene. The RNP complex was assembled by combining 2 µl of gRNA with 2 µl of diluted 10 µg/µl Cas9 protein (IDT, Cat.No.1081060), diluted in Cas 9 buffer (20 mM HEPES (stock HEPEs 1 M: Gibco, Cat.No. 15630-056);150 mM KCL (Sigma-Aldrich, P-9541), pH7.5), then incubated for 10 min at 37 °C and allowed to cool to room temperature. The gRNA was prepared by diluting 3 µl of 100 mM tracrRNA (IDT, Cat No. 1072532) and 3 µl of crRNA 5′-GAA-GAGGATTTGGATGCACT**GGG**-3′ (PAM site in bold) in 100 µl of nuclease free water, then heated at 95 °C for 5 min and allowed to cool to room temperature. Injected embryos (F0) were grown to adulthood. A male displaying a mutant phenotype (chimeric reduction of iridophores) was outcrossed to a wildtype female and the offspring (F1) were then raised. Adults of the F1 generation were swabbed and 210 bp

amplicons of exon 2 were sequenced to identify mutant carriers. The primers used were *ltk*-Fw (5'-AGTTTTGAGGAGCAGTTGTGC-3') and *ltk*-Rv (5'-TCAGTTCACTGTGGTGACTCC-3'). We identified a fish with a 10 bp deletion allele 5'-CCCAGTgcatccaaatCCTCTTC-3' (lowercase bases corresponds to deletion in WT sequence), this fish was then outcrossed to a WT fish in order to generate a stable mutant line, designated *ltk^ba7*.

## DNA extraction

1.5 ml microfuge tubes were preloaded with 50 µl of 50 mM NaOH (Fisher Scientific, Cat.No. S/4920/53) and kept on ice. Mucous samples of adult fish were taken using cotton buds and placed into individual microfuge tubes ensuring the swabs were immersed in the NaOH solution. These were then incubated at 95 °C for 15 min, afterwards the swab was placed in a 500 µl microfuge tube with the bottom cut out and this was placed back in the original tube (to create a column) and spun in a centrifuge at 2500 g for 2 min. The swab and column tube (500 µl tube) were removed and 5 µl of 1 M Tris-HCl pH 8.0 was added. These DNA solutions were then stored at −20 °C for subsequent genotyping.

## Analysis of cell fate in transient transgenic fish

The derivatives of NCCs examined were pigment cells (melanocytes, xanthophores and iridophores), glial cells (satellite glia in DRG, Schwann cells around the posterior lateral line nerve), neurons (sensory in DRG, enteric and sympathetic neurons) and jaw cartilages. All were examined sequentially in injected embryos using bright field, Nomarski or fluorescence optics as appropriate on a Nikon Eclipse E800 or Zeiss Confocal 510 META. Rohon-Beard neurons and posterior lateral line ganglia were also labeled in injected embryos. These, like other neurons, were counted after immunostaining with anti-GFP (Invitrogen, A11122)/Hu (Invitrogen, A21271) antibodies. The morphological criteria used for these cell types were as follows. 1) Melanocytes: Black pigment, dendritic form and location on neural crest migration pathways or in stereotyped locations. 2) Xanthophores: Yellowish color, granular pterinosomes, dendritic shape, and peripheral location immediately under epidermis. 3) Iridophores: localisation in dorsal, ventral (including lateral patches) or yolk sac stripes, and their round shape. 4) Satellite glia: GFP positive cells without Hu signal in DRG. DRG position was confirmed by *neurogenin1*(*ngn1*) promoter-driven red fluorescent protein (RFP) using *Tg(−8.1ngn1:RFP)*[54]. 5) Schwann cells: Elongated shape and GFP signal around the posterior lateral line nerves and spinal nerves; position of nerves was confirmed by anti-acetylated tubulin antibody (Sigma-Aldrich, T6793). Identification further confirmed by noting that GFP signal and anti-acetylated tubulin signal seemed interwoven with each other. Cell-types 4) and 5) given as glia (gli). 6) DRG neurons: *ngn1*-driven RFP in typical position next to the spinal cord. 7) Enteric neurons: Hu signal localised to gut surface. 8) Sympathetic neurons: Hu signal localised to area between ventral stripe and notochord. 9) Lower jaw cartilage: cuboidal shape and position in jaw. We counted only the cells that have RFP or Hu as neurons. Some other cells appeared likely to also be neural crest-derived neurons, such as enteric or sympathetic neurons, because of their position, but these were not included here unless marker was positive. However, neuron marker negative cells in DRG position were counted as "DRG" because they were considered to be either undifferentiated neural precursors or satellite glia, which are both neural crest-derived, but which could not be distinguished in our study. These transient transgenic studies of in vivo potency of *sox10* and *ltk*-expressing cells were performed as follows. We first scored live embryos at 2 dpf for the presence of GFP fluorescence in DRG neurons by assessing overlap with RFP signal; at 3 dpf we scored for GFP expression in melanocytes, identified by their endogenous pigment; finally at 4 dpf, we scored xanthophores by the granular appearance of their pigment granules as observed with DIC optics, and fixed the

embryos and processed them for morphological observation and for immunofluorescence using anti-GFP, anti-Hu, and anti-acetylated tubulin antibodies to confirm the presence of other NC-derived cell-types expressing GFP (iridophores, enteric neurons, sympathetic neurons, Schwann cells and cartilage).

## FACS collection of single neural crest cells

We used embryos with following genotypes for single cell profiling: (1) wild type AB Zebrafish line, (2) *Tg(Sox10:Cre)^ba74^xTg(hsp70l:loxP-dsRed-loxP −Lyn-Egfp^tud107Tg^* transgenic fish line, and (3) *Tg(Sox10:Cre)^ba74^;Tg(hsp70l:loxP-dsRed-loxP −Lyn-Egfp^tud107Tg^;sox10^m618/m618^*.

## Cell Dissociation

Embryos from required stock were grown up to the desired stage (from 14 to 72 hpf) in standard embryo media at 29 °C. To prevent melanisation in embryo melanocytes, PTU (*N*-Phenylthiourea, Sigma-Aldrich, Cat. No. P7629) was added at a final concentration of 0.003% at 24 hpf. To stimulate eGFP expression, embryos were heat-shocked by placing them in 42 °C embryo media followed by 1 h incubation at 37 °C and at least 1 h incubation at 29 °C. If required, the embryos were dechorionated using pronase (Pronase from *Streptomyces griseus*, Sigma-Aldrich, Cat. No. 000000010165921001) at a final concentration of 1 mg/ml[80]. The heads were cut from all the embryos at stage 30 hpf or older to decrease the number of *sox10*-positive cells of craniofacial skeletal and otic fates. Embryos were then digested as previously described with small modifications[81]. In brief, embryos were rinsed with Ca-, Mg- DPBS (Sigma-Aldrich, D8537), placed in a flask containing TrypLE™ Express Enzyme (ThermoFisher Scientific, Cat. No. 12605036) in ratio of 10 ml per 100 embryos, containing 0.003% Tricaine (Sigma-Aldrich, Cat. No. E10521); incubated for 30–90 min at 100 rpm, 37 °C in the shaker incubator with constant monitoring until the embryos were digested to a mixture of single cells and small fragments of tissue; then digestion mixture was triturated 10–15 times, using a Pasteur pipette; passed through 100-micron strainer (MACS SmartStrainers, Miltenyi Biotech., Cat. No. 130-098-463,) into 50 ml Falcon tube and centrifuged for 5 min, 500 x *g*, 4 °C. The cell pellet was re-suspended in DPBS and the cell suspension was passed through 30-micron strainer (MACS SmartStrainers, Miltenyi Biotech. Cat. No. 130-110-915) into 50 ml Falcon tube and centrifuged again for 5 min, 500 x *g*, 4 °C following by re-suspending the cells in 0.5–1 ml cell isolation media (2% FCS, DPBS:HBSS = 1:1 and 1 mM SYTOX Blue Dead Cell stain (ThermoFisher Scientific, S34857)). Cells were imaged before FACS and after FACS to confirm successful purification of GFP+cells.

## FACS isolation of eGFP+NCCs

Cells were excited with 488 nm laser and gated (P1) using forward and side scatter parameters to avoid debris and smallest particles. Viable SYTOX™ Blue-negative; GFP-positive cells (P2) were gated using the 405 nm laser and 450/50 nm filter (DAPI channel) and 488 nm laser and 450 nm filter (GFP/FICT channel). Cells were sorted using 488 nm laser and 530/30 nm filter (GFP/FITC channel), and the 561 nm laser and 582/15 nm filter (DsRed channel). Thresholds were established by comparison with wild type fish of the same age (not shown), and a distinct population of bright GFP-positive cells (P3; Supplementary Fig. 1a) was selected while bright red cells were excluded to avoid auto-fluorescent particles. Selected cells were confirmed by fluorescent microscopy (Supplementary Fig. 1b).

## FACS isolation of control melanocytes and iridophores

Zebrafish embryos (72 hpf) were decapitated and dissociated to a single cell suspension (Supplementary Fig. 2bi) following by centrifugation using a Percoll (Merck, Cat. No. P4937) density gradient. The cells from the bottom of the tube (Supplementary Fig. 2bii) were resuspended and sorted with Aria III cell sorter using natural cell optical properties in red (DsRed) and green (FITC-A) channels. FACS

plot (Supplementary Fig. 2a) demonstrates the relative positions of melanocyte and iridophore gating. Sorted melanocytes were selected from lower right polygon (red dots) and iridophores were selected from long upper polygon (green dots). Isolated melanocytes were imaged in bright field (Supplementary Fig. 2biii), and iridophores were imaged using both green and red channels (shown as a merged channels black-and -white image) (Supplementary Fig. 2biv).

## Single cell sorting of eGFP-positive cells into 96-well plate

Single eGFP-positive cells were sorted into each well of 96-well plate containing the lysis buffer. For cell lysis and further cDNA synthesis and Pre-Amplification we used the protocol supplied by Bio-RAD (http://www.bio-rad.com/webroot/web/pdf/lsr/literature/Bulletin_6777.pdf) with slight modifications according our primer design, and with very reproducible results. The lysis buffer was prepared on ice according to the protocol, using the whole SingleShot™ Cell Lysis Kit (SingleShot™ Cell Lysis Kit, 100 × 50 μl reactions, Bio-Rad Cat. No. 1725080) as follows: 0.8 ml SingleShot Cell Lysis Buffer, 0.1 ml Proteinase K, 0.1 DNaseI ml, 4 ml TE buffer (Sigma-Aldrich, Cat.No. 93283). Spike-in RNA (polyadenylated kanamycin mRNA, Promega, USA, Cat. No. C1381) was added to the Cell Lysis Buffer with a final concentration $10^7$ molecules/ml. 4 μl of cell lysis buffer was aliquoted into 96-well semi-skirted PCR plates on ice and immediately frozen at −80 °C. AriaIII Cell sorter was set with optimal flow parameters, the whole system cooled to 4 °C, drop delay appropriately adjusted and the precise positioning of the home device for the 96-well plate. Cells were excited with 488 nm laser and gated using forward and side scatter to avoid debris and smallest particles. SYTOX™ Blue Dead Cell stain (ThermoFisher Scientific, Cat. No. S11348) was added to final concentration of 1 uM. Viable SYTOX™ Blue-negative; GFP-positive cells (P2) were gated using the 405 nm laser and 450/50 nm filter (DAPI channel) and 488 nm laser and 450/50 nm filter (GFP/FICT channel). Cells were sorted using 488 nm laser and 450/50 nm filter (GFP/FITC channel), and the 561 nm laser and 582/15 nm filter (DsRed channel with thresholds established by comparison with stage-matched non-transgenic AB wild type fish, selecting the discrete population of bright GFP-positive cells and excluding bright red cells to avoid auto-fluorescent particles (see Supplementary Fig. 1). Cells were sorted into each well of the plate using "single cell" setup, the plate was immediately placed on ice, sealed, vortexed for 10 s, centrifuged briefly and underwent genomic DNA digestion by placing the plate in a thermocycler and incubating at 25 °C for 10 min and 75 °C for 5 min, followed by holding at 4 °C.

## Single cell sample preparation and nCounter (NanoString Technologies) quantitation of gene expression

Total RNA was converted to cDNA, using iScript™ Advanced cDNA Synthesis Kit for RT-qPCR (Bio-Rad, Cat. No. 1725038) following by 25 cycles of preamplification using mix of 47 pairs for MTE primers, each at final 50 nM concentration (Supplementary Dataset 2) and SsoAdvanced™ PreAmp Supermix (Bio-Rad, Cat. No. 1725160,). After amplification, samples were checked for quality control, and those which passed QC were selected into 12-sample strips and stored at −80 °C. When required, pre-amplified samples were thawed, hybridized with both Reporter and Capture probes, applied to the chip and chips were analysed using nCounter according to the manufacturer protocols (NanoString Technologies) https://www.nanostring.com/applicatio/files/4714/9264/4611/nCounter_XT_Assay_Manual.pdf. The NanoString data were recorded as DNA molecule counts for each gene of interest, and then were subjected to further statistical analysis.

## Quality control of the pre-amplified samples

To check absence/presence of a cell in each well of 96-well plate of sorted cells, and to check the efficiency of the preamplification steps, we utilised two custom designed Quality Control Taqman Assays: housekeeping gene *rpl13* Assay and in-RNA spike control Kanamycin

Assay. Primers and probes for both assays were designed to recognise the amplicons used for preamplification with MTE primers (Primer3 Plus software, (http://www.bioinformatics.nl/cgi-bin/primer3plus/primer3plus.cgi).

The following primers and probes were used:

| Assay | Primers | | Probe |
|---|---|---|---|
| *rpl13* | Forward | aaccagcctgccagaaagat | 5′-YAKYE-cgtcgtattgctccaagacc-3′-BHQ1 |
| | Reverse | ctcttcggccagtagtcagg | |
| *kanamycin* | Forward | gcaaaggtagcgttgccaat | 5′-FAM--atggttactcaccactgcg 3′-BHQ1 |
| | Reverse | atccccgggaaaacagcatt | |

Real time quantitative PCR was performed with TaqMan™ Fast Advanced Master Mix, (ThermoFisher Scientific, Cat. No. 4444965,) according to the manufacturer's protocol using StepOnePlus™ Real-Time PCR Systems (Applied Biosystems). Ct values were estimated using an automatic baseline and a standard threshold of 0.2. For *kanamycin*, the samples demonstrating Ct-values less than 14 were rejected as poorly amplified, for *rpl13*, the samples with Ct-values less than 24 were rejected as those with poor RNA quality or empty.

## Pigment cell enrichment

72 hpf embryos were processed as previously described[81] with minor modifications. Briefly, fish were anesthetized with Tricaine, decapitated and digested using TrypLE Express 10 ml per 100 embryos. After incubation for 45–120 min at 37 °C while shaking at 100 rpm, the suspension was triturated with a Pasteur pipette. Dissociated cells were filtered through a 300 micron strainer and immediately through a 70 micron strainer to the 50 mL Falcon tube, precipitated at 500 g for 5 min at 4 °C, resuspended in 1 mL cold isotonic Percoll (Sigma-Aldrich, Cat. No. P1644), transferred to 1.6 mL Eppendorf tubes and spun at 2000 rcf for 5 min at 4 °C in an angle rotor (Thermo Scientific, Fisher Heraeus Biofuge Primo R Centrifuge). The pellet containing pigment cells was resuspended in 400 μL of ice cold Cell Media (CM, DPBS:HBSS(no calcium, no magnesium, ThermoFisher Scientific, Cat. No. 14170112) = 1:1, supplied with 2% fetal Bovine serum (heat inactivated, ThermoFisher Scientific, Cat. No. 10082139), and placed onto preformed Percoll density gradients prepared as previously described[81]. The cells were centrifuged in a swinging bucket rotor at 2000 g for 10 min at 4 °C following by re-suspending in Cell Media and centrifugation at 500 g for 5 min. Pellet was re-suspended again in 0.5–1 ml of Cell Media, and kept on ice until Fluorescence-Activated Cell Sorting (FACS).

## Selection of melanocytes and iridophores from pigment cell enriched suspension

We utilised the natural physical properties of melanocytes and iridophores to absorb and reflect light respectively and sorted them from the suspension of pigment cells using FACS. Pigment cells enriched after Percol gradient centrifugation were resuspended in CM and filtered through 30-micron strainer (MACS SmartStrainers (30 μm), Miltenyi Biotech., Cat. No. 130-110-915). Cells were sorted with an AriaIII cell sorter using a 130-micron nozzle, in sterile conditions at +5 °C, with optimized flow parameters, drop delay and correct positioning of the 96-well plate. Cells were excited with 488 nm laser and gated using forward and side scatter to avoid debris and smallest particles. Cells were sorted based on autofluorescence using 488 nm and 561 nm filters, corresponding to DsRed and GFP channels (Supplementary Fig. 2). Iridophores were selected by high excitation in both channels, forming a scattered population in the upper right quadrant. Melanocytes do not possess any fluorescence and were detected at the lower left of the quadrant of the DsRed-FITC plot. Cells

were collected into ice-cold LB buffer in 96-well plates and immediately processed for genomic DNA degradation followed by cDNA conversion, and kept at −80 °C until next step as described above. Alternatively, cells were directly sorted to a tube containing 0.5 ml of Trizol reagent (TRIzol™ Reagent, ThermoFisher Scientific, Cat. No. 15596026) and kept on ice until mRNA extraction.

### mRNA extraction and cDNA conversion

To extract RNA from Trizol stocks we used Direct-zol™ RNA MicroPrep (ZymoResearch, USA, Cat. No. R2062) according the manufacturer's protocol. The total RNA was extracted in 10 ml, measured using Nanodrop-2000, and tested with Experion™ RNA HighSens Analysis Kits (Bio-Rad, Cat. No. 700-7105) using the Experion™ Automated Electrophoresis System (Bio-Rad). 5 ml of total RNA from either melanocytes or iridophore (50-60 ng/ml) was used directly for NanoString expression analysis.

cDNA was synthesized using 1 µl of total RNA from melanocytes or iridophores with iScript™ Advanced cDNA Synthesis Kit (Bio-Rad, Cat. No. 1725038) following by 6 cycles of pre-amplification with pooled 47 pairs of MTE primers designed for NanoString CodeSet (SsoAdvanced™ PreAmp Supermix, Bio-Rad, Cat. No. 1725160); amplified dsDNA was analysed using the NanoString expression protocol.

### Single cell quantitative RT-PCR profiling

Primers and probes for TaqMan qPCR expression assays were designed using the Primer3 web-site and ordered from Eurofins (http://www.eurofins.com/genomic-services/). Amplification primers and TaqMan assays are represented in Supplementary Dataset 3.

Zebrafish transgenic embryos at stage of 30 hpf were decapitated; eGFP-positive cells were FAC-sorted and RNA was amplified as described above with the set of 15 pairs of amplification primers (Supplementary Dataset 3). Reaction mix was diluted twice with nuclease-free water (Nuclease-Free water, Promega, Cat. No. P119C) and 1 µl of amplified dsDNA was used for gene specific TaqMan Assay using the manufacture protocol for TaqMan™ Fast Advanced Master Mix (ThermoFisher Scientific, Cat. No. 4444964). Expression data was normalized relative to sum of *kanamycin* and *rpl13* for each sample, filtered to eliminate samples with low gene expression, and transferred to R Seurat package for further analysis. Heatmaps were built with Complex-Heatmap R package. For original data, see Supplementary Dataset 4.

### Single cell data analysis

We followed general recommendations of Luecken & Theis[82] for our single cell data analysis protocol.

**The initial data**. The results of 1317 NanoString single cell transcriptomes after removal of duplicated and corrupted cells contained 1090 cells of regular, WT tail, *sox10⁻/⁻* mutant and control pigment cell types. Supplementary Table 3 contains cell type statistics for the initial set.

**Quality filtering of the cells**. To ensure the high quality of gene expression measurements we used a series of filters, which tested the data set for reliance of the control probes and internal consistency. We removed cells with poor total counts of test probes (genes) other than those of housekeeping genes (*rpl13* and *kanamycin*), assuming that most such cells related to cell types that did not express cell markers represented in our panel. In addition, we removed data for probes that displayed expression in only a small number of cells. We used the NanoStringNorm R package[83] to remove cells with poor norm factors and noisy background. Normalization was performed using the sum of probes (*kanamycin* spike-in and *rpl13* internal controls) as the reference housekeeping class with 'mean and 2sd' selected for the background. In total 731 cells (25 control iridophores, 19 control melanocytes, 108 cells from tails, 444 regular WT cells from different

stages and 135 *sox10* mutant cells) survived filtering and normalization. We filtered the expression matrix nullifying elements with values less than 30 and imputed for dropouts using drImpute[84]. In total about 20% of zero counts have been imputed into meaningful quantitative values. After imputation the log transformed expression values have been loaded into a Seurat object (ver. 2.3.4)[85].

**Descriptive statistics**. Principal component analysis demonstrated that control melanocytes, control iridophores and tail cells were shifted from the center of the main cloud in some projections (Supplementary Fig. 3). For example, melanocytes were clearly shifted along the PC2 component, with high weights of melanocyte gene markers (*mlphb, slc24a5, oca2, tyrp1b, pmela*)[81] forming a clearly separated cloud, which included also some regular cells, apparently NCC derivatives differentiating into melanocytes. In contrast *sox10* mutant cells (Supplementary Fig. 3, coloured in magenta) were not offset from the central cloud of WT cells, which is consistent with our previous suggestion that mutant cells are 'trapped' in a progenitor state (Dutton et al 2001). We observed that regular cells from different stages (hpf) did not display significant differences in their distribution, and decided to consider them as a single 'regular' class.

**Dimension reduction**. We used UMAP and tSNE, two independent algorithms for non-linear dimension reduction which conserve inter-cell distances in high dimensions (42D space of gene expressions). Cosine distance implemented in R package *proxy* was used. The control cell types were even better separated in 2D UMAP and tSNE planes than in the principle component plane; *sox10* mutants were not separated from the main cloud of WT cells overlapping with location of tail control cells but not overlapping with melanocyte and iridophore clouds (Supplementary Fig. 4a and 4b). For 2D UMAP visualization we used the following parameters: min_dist=4, spread= 9, n_neighbors=25, metric="cosine".

To ensure that the panels without and with sox10- cells (Supplementary Fig. 4a and 4b) can be compared we estimated parameters of the optimal UMAP transform using WT cells only, and then applied the resulting transform to the complete data set, with *sox10* mutant cells included.

For tSNE we used *Rtsne* R package (https://github.com/jkrijthe/Rtsne), running the optimization with the following parameters: perplexity=20, theta=0, eta=500, max_iter=50000, pca_center=FALSE, normalize=FALSE. The distance matrix for cosine distance was prepared using R package *proxy*. Both UMAP and tSNE yielded topologically similar structures with clear separation of control melanocytes and, somewhat less distinctly, control iridophores (Supplementary Fig. 5a and 5b). Tail cells also grouped together in the plot.

**DotPlots**. The DotPlot tool implemented in Seurat has expression colouring normalised independently for different genes. In the case where there are small numbers of cell types this can be visually misleading, e.g. in the Supplementary Fig. 4c expression of many weakly-expressed genes has red colouring for the sample with the highest value (e.g. contI), despite a rather modest absolute count value in this cell type, because counts in other cell types are even lower. To control this problem, we have designed our version of DotPlot (dotPlotBalanced.r) which uses a colouring scheme evaluated from expression of all genes in all cells of the panel. Our implementation is based upon ggplot2. All dotplots when it is not stated otherwise are coloured according to this scheme.

**Heatmaps**. We used the ComplexHeatmap R package (Gu et al.[86] #18299} to construct heatmaps (Fig. 2b-e and others). We used the viridis color map, for aesthetic reasons, but also to assist those with impaired colour vision. When clustering rows or column, we used cosine distance implemented in R package proxy.

**Cell clustering.** We clustered cells using as features their gene expression profiles after dimension reduction used sharing nearest neighbor clustering algorithm (Waltman et al.[87]) implemented in the Seurat toolbox (FindClusters). Since UMAP is a non-linear transform obtained by stochastic optimisation which can in some runs bring about irrelevant cluster structure, we used an enumerative algorithm to test all combinations of UMAP transform parameters and cluster resolutions (each combination of parameters was tested three times). To evaluate the clustering quality we used control melanocytes and iridophores optimising the proportion of the control cells found in the single cluster; the most efficacious heuristics maximized the following quantity, where $N_{clust\_irido}$, $N_{clust\_melo}$ are the total number of cells in the target cluster, and $N_{irido}$, $N_{melo}$ are the number of cells of the target cell type in the target cluster: $qual = \frac{1+N_{irido}}{\sqrt{1+N_{clust\_irido}}} + \frac{1+N_{melo}}{\sqrt{1+N_{clust\_melo}}}$

Clusters with similar gene expression profiles were merged using Seurat::ValidateClusters procedure based on 4 top marker genes, arriving at 7 distinct clusters (Fig. 1c, Supplementary Fig.s 4, 5). Given the stochastic nature of UMAP transformation it was encouraging to obtain robust clusters with same values of transformation/resolution parameters reproduced in replicates. Clusters containing control cell types (melanocytes and iridophores) and xanthophores (X) (identified by high expression of *pax7a*, *pax7b* marker genes) never merged with other clusters up to very high ValidateCluster threshold.

**Violin plots.** We used the standard FeaturePlot::VlnPlot -> Seurat::VlnPlot. Violin plots for the clusters are given in Supplementary Fig. 6.

**Feature plots.** We used the standard scheme implemented in the Seurat::FeaturePlot. In all cases a 2D UMAP visualization plane was used. We used the bottom cutoff on log10 gene expression (parameter FeaturePlot minCutoff = 3). Feature plots are represented in the Supplementary Fig. 7.

**Pseudotime ordering.** We used *slingshot* software[36] to construct pseudotime trajectories in the 42D gene expression space. We required trajectories to begin from eHMP and end at melanocytes and iridophores (Supplementary Fig. 9). The paucity of markers of xanthophores explains why we were unable to obtain a relevant trajectory to xanthophores. Cosine distance between clusters is not included into the *slingshot* software and was implemented from scratch.

**Comparison with published RNA-Seq data**
We selected six published datasets for single cell RNA-Seq transcriptomic profiling of *Danio rerio* developing neural crest (one dataset profiled the entire embryo; see Supplementary Table 1, containing the data for 74022 cells. The data sets are obtained for different protocols of cell marking and for different developmental phases of *Danio rerio* neural crest and its derivatives. All transcripts in the datasets were re-annotated to standard assembly Ensembl annotation. To eliminate the batch effect we used the quickCorrect function from the *batchelor* R package. Pseudo counts output by quickCorrect were normalized with Seurat package; about 19000 genes with non-zero read counts were kept. We used 35 strongest PCA components to obtain 2D UMAP maps, on which we draw feature plots. We observed that after data normalization and batch correction cells from different studies were distributed in the UMAP map mostly not by their data source but more according to the similarity of the sets of transcriptionally active genes, with cells from different data sources occupying the same areas (see Supplementary Fig. 11 and 12), which confirmed successful compensation for batch effects and that cells that are probably the same cell

type of cells remain close in the UMAP. However, normalization and batch correction often resulted in non-zero transcription activity values for cells with missing reads of the target gene, thus in Supplementary Fig. 11 we displayed the raw number of gene reads in each cell.

**Analysis of data obtained by single-cell gene expression profiling using TaqMan® gene expression assays**
**Initial data.** TaqMan RT-PCR single cell profiling data were obtained for 13 marker genes, the external spike-in (*kanamycin*), and the internal housekeeping controls (*rpl13*) in 159 cells. The list of genes was different from that used for Nanostring analysis: ***elavl3***, *ltk*, *mbpa*, *mitfa*, **neurog1**, *pax7b*, *phox2bb*, *pnp4a*, *snai1b*, *sox10*, *sox9b*, *tyrp1b*, **xdh**, the genes in bold were not present in the NanoString panel. The quantity profiled was the number of the cycles needed to obtain the threshold concentration, if the signal was not obtained in 40 cycles it was considered as 'Undetermined'. The failed experiments were marked as NA.

**Data processing and filtering.** After setting "Undetermined" elements equal to 40, the log-expressions were calculated using the formula:

$$Expression = (40 - nCycles)*\log_{10} \qquad (2)$$

The expression variance of the *rpl13* control proved to be very large (2.26), thus we used only spike-in *kanamycin* concentration for normalisation. To remove the outliers the samples with top 5% of *kanamycin* expressions were discarded. For each cell *kanamycin* expression values were subtracted from all gene expression values (as we work with log-values). Gene expressions marked as NA were set equal to zero. We removed the bottom 5% of cells regarding their overall gene expressions. Then the minimal matrix element (negative) was subtracted from each matrix elements thus transferring gene expression to positive relative units. To reduce expression noise we nullified the relative expression values less than the threshold value of 4.8. Then the matrix was supplied to the Seurat object. TaqMan data were used to build heatmaps of gene expression in samples. Supplementary Fig. 10 contains the complete heatmap of all TaqMan assay probes. A heatmap of probes with positive *ltk* (22 probes in total) shows that *ltk* is indeed co-expressed with markers of different cell fate (Fig. 2d).

**Statistics and reproducibility**
Statistical testing was by two-tailed *t*-test (Supplementary Fig. 17) or Chi-squared test (analysis of data in Table 1). For details of statistics and reproducibility, see Source Data file.

**Reporting summary**
Further information on research design is available in the Nature Portfolio Reporting Summary linked to this article.

## Data availability
The NanoString nCounter raw data and TaqMan assay data have been deposited in the National Center for Biotechnology Information Gene Expression Omnibus (GEO) and are accessible through the GEO Series accession number "GSE185592". All other relevant data supporting the key findings of this study are available within the article and its Supplementary Information files or from the corresponding author upon reasonable request. Source data are provided with this paper.

## Code availability
All custom scripts have been deposited at https://github.com/SevaVigg/NanoStringDanioNCCscAnalysis and are available in Zenodo with the identifier https://doi.org/10.5281/zenodo.7585731 [88], distributed under MIT licence.

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

## Acknowledgements

The authors gratefully acknowledge the Technical staff within the Department of Biology & Biochemistry at the University of Bath for technical support and assistance; Nathaniel S. Gray for providing us with ALK inhibitor, TAE684; Richard Jäger for *p80* cDNA for template of *NPM*, reagents for BAC recombineering from Biological Resources Branch of Frederick National Library for Cancer Research, and Shin-ichi Higashijima for *egfp-polyA* cassette for recombineering. Rosalind John (University of Cardiff) kindly supplied the reagents for the RecA mediated PAC recombineering. We thank Marie Srotyr and Tia Dally for their technical assistance in experiments for Supplementary Fig. 11a. We thank Alfonso Martinez-Arias for helpful discussions in the early years of this project. We thank Adele Murrell and Andrew Ward for their critical comments on an earlier draft of this manuscript. This work was supported by Uehara Memorial Foundation (MN), Wellcome Trust VIP awards (M.N.), and BBSRC grants BB/ L00769X/1(R.N.K., H.S., T.S.) and BB/S015906/1 (R.N.K., J.H.P.D., K.C.S., G.B.) and BB/ L007789/1 and BB/S01604X/1 (A.R.), National Natural Science Foundation of China, Grant Number: 31000542 (X.Y.), Royal Society International Exchange Cost Share 2017 Russia award (R.N.K.), Russian Foundation of Basic Researcher grant 17-54-10014 (V.J.M.), Ministry of Science and Higher Education of the Russian Federation Grant number 075-15-2021-601 (V.J.M.), and University of Bath PhD Studentship and ORS award (T.J.C.).

## Author contributions

R.N.K., H.S., J.H.P.D. and A.R. conceived and designed the study and obtained funding. M.N., T.S., K.C.S., G.B., X.Y., F.S.L.M.R., N.S. and T.J.C. performed the experimental studies and analysed and interpreted the data obtained. A.S.K. and V.J.M. designed the pipeline for Nanostring data processing, L.A.U. and V.J.M. wrote the code. R.N.K. and V.J.M. drafted the manuscript, and all authors contributed to revision of the manuscript. All authors have approved the submitted version.

## Competing interests

The authors declare no competing interests.
