## [Peer Review File · Nature Communications]

Zebrafish pigment cells develop directly from persistent highly multipotent progenitorsREVIEWER COMMENTS

Reviewer #1 (Remarks to the Author):

The paper by Nikaido and Subkhankulova et al examine different models of neural crest cell fate determination in zebrafish. Using Nanostring sequencing, expression, overexpression and fate mapping they argue for a “new” model of broadly multipotent neural crest cell state that undergoes a cyclical model of fate determination. While the paper adds to the existing information about neural crest cell fate, the data are not novel nor convincing to support this cyclical fate restriction model. My comments are below:

Major issues:

1. As currently written, it is not clear how this data support a fundamental different mechanism of neural crest cell fate specification than what has been published previously. The authors argue that there is still a controversy over whether neural crest cells have a more restrictive progenitor but several lines of evidence both from more classic lineage studies in multiple species (Baggiolini et al, Bronner lab papers and others) as well as more recent data suggest that this is not the case. There have been many recent single cell RNA sequencing papers that have focused on neural crest cells in multiple species (Soldatov et al, Farrell et al, Howard et al, Lencer et al) as well as nice quantitative expression analysis (Roellig et al) that supports a model of cell fate determination where cells remain multipotent and express markers of multiple derivatives before committing to one lineage or another. Roellig, the most similar to the data presented here, emphasize that NCC progenitors express genes associated with multiple fates at least transiently using high resolution in situ expression analysis. However, it is unclear how do these data presented here differ from a progressive or even direct model of differentiation or how these data suggest that cells transition from one state to another then back in a cyclic form, if that is what is being argued? If truly a new model, this should be more clearly discussed with more specificity as to what is proposed and how this may differ from classic ideas of cell fate. A longer discussion engaging with prior research may do much to strengthen the authors claim. As is, the reader is left unsure of what the cyclic model is and thus cannot be fully convinced by how the data leads to this different way of thinking about cell fate.

2. This leads to another issue which is that the Nanostring method used only screened for 45 genes in overall very few cells (1317), as opposed to a single cell whole transcriptomic approach such as 10xgenomics single cell (sc) RNA-sequencing. The authors do address how their dataset samples genes associated with differentiated cell types, but a longer discussion is warranted. Does Nanostring allow deeper sequencing to identify low expressed genes and thus have less dropout than other platforms? If so Nanostring may be better at picking up such mixed fate cells and contrast with scRNA-seq datasets which would require a lot more depth in order to capture these transient events. In contrast, one wonders at what point low expression is biologically significant? On the other hand, are biologically

important cell clusters or populations of these actively differentiating neural crest cells being missed because only a limited gene set is being investigated? The authors have evidently considered these issues but more discussion in the manuscript is warranted given the now widespread use of different single cell genomic technologies.

3. Related to this, in selecting a limited set of genes is that these genes might be good markers for different cell types at later stages, but they may be poor markers at the pre-migratory stages discussed here. For instance perhaps *Ltk* is just not an early marker of iridophore cell type? The risk is that conclusions are drawn assuming a gene as a marker for a cell type when in fact the gene is broadly expressed and then refined.

4. Another concern, it is very confusing as written what is considered control cells in this context. The use of abbreviations and the way the figures are labeled makes it very difficult to understand. Why were wildtype tail cells and *sox10* mutant cells used as “controls”? This needs to be explained. In addition, as stated above, the number of cells sequenced over this broad time frame is very limited and within that there are two large populations of cells, cluster 4 and 7, are not really discussed as to what they represent. These represent over half the data set. This does not give confidence that the authors have addressed all their data in a rigorous way.

5. The functional work and cell lineage tracing data argue against the concern that *Ltk* is not expressed in early neural crest lineage. But these data could be strengthened. Lacking are convincing whole mount in situ expression analysis of *Ltk* at different stages and at high magnification confirming that the transgenic *Ltk:GFP* line truthfully replicates endogenous expression. This is important for any work using this line for lineage tracing. In situ showing cell expressing genes associated with multiple fates are convincing but there is no quantification in a rigorous way or discussion which would strengthen these data. Do all NCCs express *Ltk* and *phox2b*? It seems not. Thus what percentage of cells co-express or singly express these genes? Or, is there a quantitative relationship between the level *Ltk* and *phox2b* expression in single cells?

6. The *Ltk/sox10* overexpression experiments are informative and a nice addition. However, these experiments would be more informative to the argument of ‘cyclic cell fate’ if the authors looked at the expression of different markers in these overexpressed cells? Additionally, it would be important to use markers to show that iridophore fate is promoted at the expense of melanocytes and vice versa in the same embryo. While there is some quantification in Table 1, it is very limited and needs to be expanded. The data in Extended Data Figure 9 seems critically important to the argument of bimodal *Ltk* function, which makes it odd that these data are in the supplement. The authors may want to consider putting these data into the main text. If available better images may strengthen these data as well. Importantly, what happens in the *Ltk* mutant?

More issues:

1. Figure 1 should include the cyclical fate model and how it relates and is different than the existing models of cell fate determination.
2. The figures abbreviations make reading and understanding the data very difficult. The tables have numbers but only *gfp+* information. Table 1 has “*gfp+* embryos observed in embryos”, what does this mean?
3. In the melanocyte cluster, it is not clear if these are real differences since there is a limited number of marker genes are used in Extended Figure 5, there is a differentiation from left to right, but again there are a lot of the cluster 7 cells in there, what are these?
4. Thus, the pseudotime analysis is confusing as displayed.

Reviewer #2 (Remarks to the Author):

The manuscript by Nikaido and coauthors is a nice study aiming at the multipotency and fate restrictions in fish neural crest cells with in depth focus on the development of pigment cell repertoire. Neural crest is a highly multipotent population of progenitor cells, which transiently exist during vertebrate embryonic development. Neural crest cells give rise to dozens of cell types and are responsible for the development of autonomic and sensory nervous systems, as well as facial and cardiac morphogenesis. The enigma of the neural crest multipotency and the mechanisms partitioning the downstream fates is still on top of the list. Here, Nikaido and coauthors attempt to address the problem of a cell fate choice in the fish neural crest lineage (looking mostly at pigment cell progenitors) using a single cell technology, some functional tests and lineage tracing. Although limited, the data are strong and validated, and I feel enthusiastic and rather support the publication of this study after addressing the most critical issues.

Below I list the major concerns:

1. It would be beneficial to obtain regular single cell transcriptomics datasets for the unbiased analysis of the neural crest populations. Clustering and trajectory inference based on few preselected genes is not reliable and can be used only to address a narrow scope of questions. At the same time, the authors performed validations of gene expression, including a co-expression analysis, which makes their observations more solid. The main figure does not show the trajectory and co-expression of opposite programs/master regulators in the same progenitor cells. The mentioned trajectory is missing from the Extended Data Figure 5. Navigating such trajectory must be super hard and unreliable given these sparse data. This is a very important point of the study and should be brought up in the main figure (Figure 2), preferably after obtaining a regular 10x or Smartseq-based dataset.

2. The authors write: “This provides a strong indication of the highly dynamic nature of the fate specification and differentiation process – that apparently specified cells identified *in vivo* are held in states biased towards one or more fates by environmental signals.”

I believe the authors do not sufficiently discuss the essential progress made by other teams in this direction. The states of biased NCCs were carefully observed and experimentally tested in mice by Soldatov et al. 2019, which the authors cite to discuss another set of arguments. The combination of opposite genetic programs in the same individual neural crest cells was also observed by these authors. The results reported in this manuscript go perfectly well along these discoveries in mouse neural crest, and the authors shall discuss this convergence in data and ideas (co-activation and competition of programs in same cells), including the evolutionary aspect (conservation of this type of fate conflicts in fish and amniote neural crest).

3. Next, the authors should be careful with their conclusions as they do not test the actual fate restrictions, and instead they observe the transcriptional landscape and possible transcriptional conflicts. Although the cells show the co-expression of opposite transcriptional programs and individual master genes, they might be already fate restricted in one specific direction depending on the level of program competition, the epigenetic state, the signaling landscape and many other factors. The authors need to separate the genealogical portrait (actual situation) from the transcriptional portrait, which is merely “instrumental”. The genealogical portrait can be obtained only experimentally via different types of clonal lineage tracing and perturbation experiments, and the concept of progressive or any other type of fate restrictions can be applied only to a genealogical portrait of the system. The transcriptional portrait, however, is useful to predict and understand the mechanisms providing the genealogical outcomes. This theory of genealogical vs descriptive transcriptional analysis must be extensively discussed throughout the manuscript, and the authors should interpret their data in a context of this accurate theory. This is exactly why I welcome clonal tracing and more of the functional experiments to test the cyclic fate restrictions model proposed by the authors. Without extensive functional experiments and quantitative clonal analysis of fate restrictions, the concept of CFR does not bear any significant weight beyond being a hypothesis (which is not so different from PFR concept in such unsupported case). I would really love to see the clonal data, with probability distributions, as they would make the author’s point ultimately strong (some inspiration can be obtained from the reference number 56: Singh et al., *Dev Cell* 2016: [https://www.cell.com/developmental-cell/pdf/S1534-5807\(16\)30423-3.pdf](https://www.cell.com/developmental-cell/pdf/S1534-5807(16)30423-3.pdf)).

The model of CFR must be at least reconciled with the existing clonal genealogical data for a zebrafish pigment cell lineage reported by Singh et al., *Dev Cell* 2016. The transcriptional portraits shall be interpreted in this context, and possible explanations need to be provided for any identified inconsistencies.

4. The authors performed fate-mapping experiments from Ltk-expressing cells and found multiple progeny beyond pigment cell lineage. This suggests that the expression of Ltk is not restricting the potential of a cell lineage. At the same time, the expression of Ltk can still be a passive or active biasing factor, and this should be addressed. I wonder if the early progenitor cells expressing Ltk end up in specific pigment cell sublineage more often as compared to the cells not expressing Ltk at the early stage (in clonal analysis). The authors should work with probabilities of different outcomes depending on the early expression of a given biasing factor.

5. Again, a systematic clonal analysis (fate distribution in single clones traced from a biasing factor / master regulator) will be highly instrumental.

6. Can the authors use the power of a zebrafish system to bias or re-bias NCCs in the context of competing programs?

Again, I feel enthusiastic about this story and hope to see it again after all improvements.

Reviewer #3 (Remarks to the Author):

Here, Nikaido and Subkhankulova, et al uncover a cyclical fate restriction mechanism that describes the development of zebrafish pigment cells from a multipotent neural crest intermediate state into differentiated cell states. Using single cell Nanostring profiling, they were unable to recover a partially restricted intermediate population suggesting fate restriction arises from a highly multipotent intermediate population. Further, the authors investigate the biphasic expression of *ltk*, which marks multipotent neural crest as well as differentiated iridophores. By inhibiting or activating *Ltk*, the authors test the multipotency of *ltk*-expressing cells. This manuscript presents a new model for fate specification of pigment cells, which should be investigated in other neural crest cell types in the future. The following concerns should be addressed prior to publication:

Main:

1. With such a limited gene set for Nanostring (45 genes), the scope of understanding the fate restriction mechanism is limited to the genes we already know. How were the 45 genes chosen for the probe set?

2. (Line 261) The number of iridophores should be quantified as well. Were there differences in the number of neurons after altered *Ltk*?

3. How does constitutive LTK affect neurons and glia?

4. Can the authors comment on how Schwann Cell Precursors may confound their results and cyclical model as they are known to remain multipotent later into development.

5. Fig2: More stages should be investigated with FISH

Minor:

1. FAC-sorted, not FACS sorted

2. The authors mention 8 time points were isolated (Line 84), what are these time points? Only 7 are shown in Fig2.

We thank the referees for their careful consideration of our manuscript and are grateful for the opportunity to respond to their comments. Our responses are given in red below.

REVIEWER COMMENTS

Reviewer #1 (Remarks to the Author):

The paper by Nikaido and Subkhankulova et al examine different models of neural crest cell fate determination in zebrafish. Using Nanostring sequencing, expression, overexpression and fate mapping they argue for a “new” model of broadly multipotent neural crest cell state that undergoes a cyclical model of fate determination. While the paper adds to the existing information about neural crest cell fate, the data are not novel nor convincing to support this cyclical fate restriction model. My comments are below:

Major issues:

1. As currently written, it is not clear how this data support a fundamental different mechanism of neural crest cell fate specification than what has been published previously. The authors argue that there is still a controversy over whether neural crest cells have a more restrictive progenitor but several lines of evidence both from more classic lineage studies in multiple species (Baggiolini et al, Bronner lab papers and others) as well as more recent data suggest that this is not the case. There have been many recent single cell RNA sequencing papers that have focused on neural crest cells in multiple species (Soldatov et al, Farrell et al, Howard et al, Lencer et al) as well as nice quantitative expression analysis (Roellig et al) that supports a model of cell fate determination where cells remain multipotent and express markers of multiple derivatives before committing to one lineage or another. Roellig, the most similar to the data presented here, emphasizes that NCC progenitors express genes associated with multiple fates at least transiently using high resolution in situ expression analysis. However, it is unclear how do these data presented here differ from a progressive or even direct model of differentiation or how these data suggest that cells transition from one state to another then back in a cyclic form, if that is what is being argued? If truly a new model, this should be more clearly discussed with more specificity as to what is proposed and how this may differ from classic ideas of cell fate. A longer discussion engaging with prior research may do much to strengthen the authors claim. As is, the reader is left unsure of what the cyclic model is and thus cannot be fully convinced by how the data leads to this different way of thinking about cell fate.

We thank the reviewer for the opportunity to expand our arguments as to the incompatibility of our data with the previous models. It is important to be clear that our model is not a new view of ‘fate specification’, but is specifically a new model of ‘fate restriction’ (we have noted the potentially misleading reference to ‘specification’ in the Abstract, and have now edited this to ‘restriction’. We have also taken the opportunity to thoroughly review our use of such terminology to ensure clarity). The key issue concerns the potency (*not* specification state) of individual neural crest cells at different stages of pigment cell differentiation, a debate that goes back to some of the foundational contributions to the field by Bronner, Le Douarin, Weston, Anderson and others (see Kelsh et al., 2021; Dawes and Kelsh, 2021). As we explain in detail in our recent reviews, the previous studies do not provide a definitive answer to the question of multipotency: for example, the Baggiolini et al study claims to show multipotency, but uses a definition of multipotency of ‘more than 1’ fate, which does not resolve the difference between the Direct Fate Restriction (DFR) and Progressive Fate Restriction (PFR) models. In line with the reviewer’s suggestion, we add an explicit, but

necessarily brief, reference to this and other recent discussion in the introduction to help set the context for our work. A full description of our new CFR model has now been published (Kelsh et al., 2021; Dawes and Kelsh, 2021), and these provide a critique of the current models, as well as explaining how the CFR model provides an explanation for the original data underpinning those DFR and PFR models. We have edited the Abstract and Discussion (see penultimate paragraph) to clarify this and to explain how the CFR model is consistent with our new data.

The reviewer mentions the paper by Roellig et al. as being most similar to ours. We presume the reference is to Roellig et al. (2017, eLife.21620.001), although this paper addresses gene expression and cell fate analysis in the neural plate border, and thus targets an earlier stage and different process (NC induction from ectodermal cells, choice between epidermis, neural tube, placode and neural crest). However, its demonstration of expression of a broad range of fate-specific marker genes in neural plate border cells is analogous to what we are suggesting here in the neural crest. We now include reference to this in the final sentence of the Discussion, and highlight too another example.

The single cell studies from the zebrafish which focus on pigment cell development (Howard et al., Lencer et al., and others), largely published in the time whilst our manuscript has been in submission, are all interpreted in a PFR context and clearly argue against a DFR model. However, as noted below, we believe that a critical examination of those data leaves them unable to resolve the issue we are addressing here.

The inadequacy of the current DFR and PFR models is shown by the following data in our study:

- A) Isolated cells from even surprisingly late stages co-express markers of a much broader range of lineages when assessed by the very sensitive Nanostring and Taqman techniques. We interpret co-expression of such markers as transcriptional evidence for *minimal potency* i.e. if a cell is able to co-express such key factors for two or more fates then it must retain a minimum potency that includes those two or more fates. The published papers that the reviewer references are based on scRNA-seq and indicate the pathways of differentiation; as the reviewer states, these studies document cells passing through states in which they co-express markers of multiple (usually two e.g. Soldatov et al., 2019) lineages (i.e. are co-specified to multiple lineages). We note that the zebrafish scRNA-seq studies mentioned often lack the sensitivity to detect the key fate specification factors (transcription factors (e.g. *mitfa*, *phox2bb*) and fate specification receptors (e.g. *ltk*)), as we now show in our own meta-analysis of these data from studies of zebrafish neural crest (new Extended Data Figures 7-9). Thus, whilst the scRNA-seq studies nicely document changes in fate specification and differentiation, we show that they underestimate fate potential in such cells.
- B) Our use of RNAscope *in situ* hybridisation to assess the coexpression of subsets of these markers reinforces this view, especially since it reveals overlaps *in vivo* that are very unexpected under the PFR view (e.g. *phox2bb* and *ltk* in cells associated with the horizontal myoseptum, i.e. posterior lateral line nerve glial cells, a location where neither sympathetic neural nor iridophore cell-fates are found). In this revised manuscript, we have expanded these observations (see new Extended Data Fig. 10) and their discussion (see details below).
- C) In an independent demonstration of the broad multipotency of apparently specified neural crest cells, we show that *ltk*-expressing cells in the early expression phase (predominantly premigratory neural crest cells) retain broad potency that includes all pigment cell, glial and neuronal fates. Under the PFR model, as proposed in our earlier study (Lopes et al., 2008;

- Petratou et al, 2018,2021), *ltk* might be expected to mark chromatoblasts, progenitors restricted to all pigment cell fates, but certainly not to include neuronal and glial fates too.
- D) We build on our original demonstration that constitutive activation of Ltk signaling in the neural crest drives cells to form supernumerary and ectopic iridophores, by now providing data that these cells are excluded from becoming melanocytes, and that this reflects an early repression of expression of the melanocyte master regulator gene, *mitfa* (Extended Data Fig. 15). Further we also provide new data indicating that this repression extends to *phox2bb* (enteric and sympathetic neural fates) and *pax7b* (xanthophores) (Extended Data Fig. 15).
 - E) We now add a meta-analysis of the recently published zebrafish scRNA-seq studies of pigment cell development (section entitled 'Nanostring hybridization profiling generally agrees with published scRNA-seq data but can identify genes with low expression'), which makes clear that these studies give quite a detailed description of the transcriptional profiles forming during differentiation, but that the depth of sequencing routinely obtained is insufficient to give a readout of potency, and so they do not distinguish between the PFR and CFR models (new Extended Data Figures 7-9).

We have now expanded our discussion of our data to make these points more explicit.

2. This leads to another issue which is that the Nanostring method used only screened for 45 genes in overall very few cells (1317), as opposed to a single cell whole transcriptomic approach such as 10xgenomics single cell (sc) RNA-sequencing. The authors do address how their dataset samples genes associated with differentiated cell types, but a longer discussion is warranted. Does Nanostring allow deeper sequencing to identify low expressed genes and thus have less dropout than other platforms? If so Nanostring may be better at picking up such mixed fate cells and contrast with scRNA-seq datasets which would require a lot more depth in order to capture these transient events. In contrast, one wonders at what point low expression is biologically significant? On the other hand, are biologically important cell clusters or populations of these actively differentiating neural crest cells being missed because only a limited gene set is being investigated? The authors have evidently considered these issues but more discussion in the manuscript is warranted given the now widespread use of different single cell genomic technologies.

The reviewer highlights what is for us a key point – it is clear that the Nanostring and Taqman assays gave greater sensitivity than is usually achieved with scRNA-seq. This is best shown by our meta-analysis of the recent scRNA-seq studies of zebrafish pigment cell development, in which many of our key genes are not readily detected in progenitor cells (despite published ISH studies showing early expression (e.g. *ltk*)), but also by our RNAscope assays which also give very high sensitivity. The significance of low level expression is likely to vary with context: in the dynamic view proposed here, low levels may simply be transient phases, or may reflect cells taking different differentiation paths and thus turning off the irrelevant gene. These low levels may or may not have functional consequences in themselves, but given the known biological roles of all the genes emphasised in our studies (including *ltk*, *mitfa*, *tfec*) we are confident that the very fact of their being expressed has significance in indicating that the cells have, or recently had, potential to adopt that specific fate(s). In contrast, the broad detection of thousands of genes, but biased towards those expressed at high levels, makes scRNA-seq very good at detecting transcriptional profiles of cells undergoing differentiation, but less sensitive to the low level expression that would reflect cell potency. We have expanded the discussion of this point, as noted in the response above. Specifically, we have included a new section *Nanostring hybridization profiling generally agrees with published RNA-Seq*

data but can identify genes with low expression, in which we conduct an integrative study of several RNA-Seq datasets, and compare gene expression profiles identified with RNA-Seq with those identified with Nanostring.

3. Related to this, in selecting a limited set of genes is that these genes might be good markers for different cell types at later stages, but they may be poor markers at the pre-migratory stages discussed here. For instance perhaps *Ltk* is just not an early marker of iridophore cell type? The risk is that conclusions are drawn assuming a gene as a marker for a cell type when in fact the gene is broadly expressed and then refined.

We acknowledge the reviewer's point is in general a valid one, but that is exactly why we focused our study on genes already shown to be biologically necessary for adoption of the key fates described. Thus, for example, in both *ltk* and *tfec* mutants it has been shown that NCCs fail to become specified as iridoblasts even from very early stages (Lopes et al., 2008; Petratou et al., 2021), and *mitfa* is widely acknowledged as a master regulator of melanocyte fate, in mammals and zebrafish (Steingrimsson et al., 1994; Lister et al., 1999). Furthermore, we have now expanded our study of the impact of constitutively-activated *Ltk* signalling, showing now that this results in an early repression of non-iridophore fate specification transcription factors (*mitfa*, *pax7* and *phox2bb*)(new Extended Data Figure 15). These data clearly show that the genes we highlight really are key genes driving fate choices, and hence are *the* most suitable as markers of fate potential. We have edited the sentence introducing these in the paragraph beginning 'We investigated the transcriptional profiles of...' and the Discussion paragraph beginning 'We have been unable to identify clusters ...' to clarify this point.

4. Another concern, it is very confusing as written what is considered control cells in this context. The use of abbreviations and the way the figures are labeled makes it very difficult to understand. Why were wildtype tail cells and *sox10* mutant cells used as "controls"? This needs to be explained. In addition, as stated above, the number of cells sequenced over this broad time frame is very limited and within that there are two large populations of cells, cluster 4 and 7, are not really discussed as to what they represent. These represent over half the data set. This does not give confidence that the authors have addressed all their data in a rigorous way.

We regret the use of the word 'control' here, since it has clearly generated confusion, although our intention was simply to highlight sets of neural crest cells which were at least partially-defined. The control melanocytes and iridophores are differentiated pigment cells sorted directly using a published gradient-density centrifugation protocol, and validated by their expression of pigment (here we believe the word 'control' was appropriate). The WT tail cells from 24 hpf embryos are 'controls' for early stages, simply because the neural crest at this stage in this region of the embryo is in the early stages of expression of *sox10*, and there is little migration as yet; these cells thus help to independently define early stages (Kelsh, 2006). The status of neural crest cells in *sox10* mutants was investigated in the context of our studies of iridophore development, and provided a unique opportunity. Our previous work had shown that progenitors for pigment cell fates failed to migrate in *sox10* mutants, instead being trapped in a premigratory position (Dutton et al., 2001); these premigratory neural crest cells are thus enriched in *sox10* mutants, and have been tentatively identified as candidate chromatoblasts, based on retained expression of *ltk*, *tfec* (but not early neural crest markers; Lopes et al., 2008; Petratou et al., 2018, 2021). Regardless of whether or not they are actually chromatoblasts (a hypothesis disproved by our studies here), they are clearly

trapped in an early stage of neural crest development, somewhere between initial 'early neural crest' and specified pigment cell progenitors. To help clarify these points we have:

- A) Edited paragraph beginning 'We investigated the transcriptional profiles...' to remove reference to WT tail and *sox10* mutant cells as 'controls'
- B) Edited Methods and extended data figures accordingly. Thus, a) In the caption to Extended Figure 1 we have modified our notation, reserving the term 'control' for only the control melanocytes and control iridophores; b) in the legend to Extended Figure 5 we have added 'Cells from cluster 7 express a limited number of differentiation markers and their identity is uncertain.'
- C) In the main text, we have added the paragraph:
'Amongst those cells included within the pseudotime ordering of melanocyte and iridophore development are a few belonging to cluster 7 (Extended Data Fig. 5), which we were unable to assign to a particular cell type with the selection of marker genes at hand. Extended Data Fig. 1d shows that cluster 7 displays *id2a* expression, which is common in all NCC derivatives, as well as that of some other genes visible at early stages of NCC differentiation (*alx4b*, *ednrba*, *her9*, *hmx4*, *impdh1b*, *mc1r*, *pax7b* and *tfap2a*). This cluster does not express any specific markers of pigment cells, such as *mlphb* and *oca2* (melanocyte), *pnp4a* (principally iridophores, but at lower levels in melanocytes and xanthophores) or *pax7a* (xanthophore) cell types. With the set of tested markers we are unable to assign the cells of cluster 7 to any of the other cell types, although they presumably include various fate-biased cell-types distinguished by marker genes not included in our gene set. Cluster 4 lacks most genes in the Nanostring gene-set, except *foxp4*, *her9*, *id2a*, and *impdh1b*, with some expression of the early *tfap2a* gene.'
And on the next page, we add the following: 'Interestingly the expression profile of *sox10* mutant cells is also close to that of cluster 7, except that expression of various genes, including *snai1b* and *sox10*, is elevated in *sox10* mutant cells, again perhaps indicating that they are trapped in an early progenitor state of some form.'
- D) And in Extended Data Figure 3 we add 'The identity of cluster 4 is less clear, but in the zebrafish RNA-Seq data the intersecting area of the genes defining this cluster (*foxp4*, *her9*, *id2a*, and *impdh1b*, with some expression of the early *tfap2a* gene) more or less overlaps with the area of expression of *twist1a* (see Extended Data Figures 7 and 9), so that at least some of these cells may include skeletogenic fates.'

5. The functional work and cell lineage tracing data argue against the concern that *Ltk* is not expressed in early neural crest lineage. But these data could be strengthened. Lacking are convincing whole mount *in situ* expression analysis of *Ltk* at different stages and at high magnification confirming that the transgenic *Ltk:GFP* line truthfully replicates endogenous expression. This is important for any work using this line for lineage tracing. *In situ* showing cell expressing genes associated with multiple fates are convincing but there is no quantification in a rigorous way or discussion which would strengthen these data. Do all NCCs express *Ltk* and *phox2b*? It seems not. Thus what percentage of cells co-express or singly express these genes? Or, is there a quantitative relationship between the level *Ltk* and *phox2b* expression in single cells?

We have previously published an *in situ* hybridisation time-course of *Ltk*, but took the opportunity here to repeat this with our more sensitive RNAscope methodology. This data is now presented in new Extended Data Figures 12 and 13) and the accompanying text, referenced in the section 'Biphasic expression of *Ltk* distinguishes....'.

We also addressed the question of faithful reproduction of this *ltk* pattern in the *ltk:GFP* line by combining GFP detection with *ltk* RNAscope in mosaic embryos injected with the *ltk:GFP* construct (new Extended Data Fig. 16)

We now present detailed quantification of the *ltk* and *phox2bb* co-expression data in a new Extended Data Fig. 11, referenced in the section 'Co-expression of key fate specification genes in situ'. We provide counts for cells expressing *ltk* in premigratory NCCs of 24 hpf and iridophores at 2dpf (Extended Data Figure 13c-e), when essentially all such cells express *ltk*. In contrast, the numbers expressing *ltk* and/or *phox2bb* at later stages and more widely in the embryo is much lower. Hence, the percentage of cells doubly expressing these genes is low, but we were unable to provide quantitation of the percentage due to the sheer number of NCCs and difficulties in distinguishing the boundaries of adjacent cells (and thus counting them) in locations such as the nerves. Instead we provide quantitation of *ltk* and *phox2bb* expression in *doubly-positive* cells (Extended Data Figure 11a). The question of the quantitative correlation between these genes is an interesting one; our data (Extended Data Figure 11a) indicates that there is wide variation in the absolute numbers of transcripts and in the ratios of the two transcripts in these cells.

6. The *Ltk/sox10* overexpression experiments are informative and a nice addition. However, these experiments would be more informative to the argument of 'cyclic cell fate' if the authors looked at the expression of different markers in these overexpressed cells? Additionally, it would be important to use markers to show that iridophore fate is promoted at the expense of melanocytes and vice versa in the same embryo. While there is some quantification in Table 1, it is very limited and needs to be expanded. The data in Extended Data Figure 9 seems critically important to the argument of bimodal *Ltk* function, which makes it odd that these data are in the supplement. The authors may want to consider putting these data into the main text. If available better images may strengthen these data as well. Importantly, what happens in the *Ltk* mutant?

As suggested, we have extended our studies of the effects of constitutive activation of *Ltk* signaling, looking at the expression of different markers. We have shown that injection of *Tg(sox10:NPM-Ltk)*, but not *Tg(sox10:NPM-Ltk(DK))* is incompatible with differentiation of melanocytes, but now we have addressed the prediction that activated *Ltk* signalling will drive the adoption of iridophore fate by activating *tfec* expression (new Extended Data Fig. 15). Furthermore, we now show that the repression of melanocyte development acts at an early stage, by repressing expression of *mitfa* itself (new Extended Data Fig. 15). We extend this to show that activated *Ltk* signalling also represses key transcription factors mediating xanthophore (*pax7b*) and autonomic neurons (*phox2bb*) (new Extended Data Fig. 15). This greatly extends the evidence that *Ltk* signalling in early NC actively drives iridophore fate, whilst repressing other fates.

We thank the reviewer for the suggestion regarding moving the data on biphasic action of *Ltk* from Extended Data Fig. 9 to the main text: it is now shown in Fig. 3a-m, but with the original images replaced by better quality ones and supplemented with further quantitation. We have also performed the experiment to do the inhibitor treatment in strong *ltk* mutants as suggested by the reviewer. To test the effect of the *Ltk* inhibitor on this phenotype necessitated us generating a new null allele (*ltk*^{Δ10bp}), since the original allele (*ty82*) had been lost during COVID pandemic. As expected, iridophores remained absent in these mutants in the presence of the inhibitor. This is now shown in Fig. 3j-m. We have also taken the opportunity to add 2 sentences to the section 'Biphasic expression of *ltk*...' to summarise the background information on *Ltk* function from our previous work to clarify this section.

More issues:

1. Figure 1 should include the cyclical fate model and how it relates and is different than the existing models of cell fate determination.

We have considered carefully this suggestion, but have decided to retain the introduction of CFR later in the paper, principally because it is the data presented here that allow us to reject the PFR model and require us to rethink mechanisms. Consequently, we have adjusted the flow in the Abstract to reflect this too. In addition, we have made reference to our published descriptions of the model and its justifications in the Discussion and have expanded our description somewhat to clarify.

2. The figures abbreviations make reading and understanding the data very difficult. The tables have numbers but only gfp+ information. Table 1 has “gfp+ embryos observed in embryos”, what does this mean?

In Table 1, “gfp+ embryos observed in embryos” refers to numbers of embryos from total examined that actually show GFP expression. We have now changed this to ‘embryos showing GFP-positive cells’ for clarity.

We have now removed abbreviations from Table 2.

We have assessed all figures and in the text for those where abbreviations are unhelpful. In the caption to Figure 2 and Extended Figure 1 we have now spelt out and explained the notation used, reserving the term 'control' for only the control melanocytes and control iridophores; we also checked that all abbreviations are spelt out in the text.

3. In the melanocyte cluster, it is not clear if these are real differences since there is a limited number of marker genes are used in Extended Figure 5, there is a differentiation from left to right, but again there are a lot of the cluster 7 cells in there, what are these?

We have added a paragraph annotating cluster 7 into the main text (see reply to remark 4 of your review above). Additionally, we have added a paragraph on cluster 4: ‘Cluster 4 lacks most genes in the Nanostring gene-set, except *foxp4*, *her9*, *id2a*, and *impdh1b*, with some expression of the early *tfap2a* gene; in the scRNA-Seq data the intersecting area of these genes more or less overlaps with the area of expression of *twist1a* (see Extended Data Fig. 8 and 9), so that at least some of these cells may include skeletogenic fates.’

4. Thus, the pseudotime analysis is confusing as displayed.

We have added a corresponding paragraph in the text. See the reply to remark 4 of your review above.

Reviewer #2 (Remarks to the Author):

The manuscript by Nikaido and coauthors is a nice study aiming at the multipotency and fate restrictions in fish neural crest cells with in depth focus on the development of pigment cell repertoire. Neural crest is a highly multipotent population of progenitor cells, which transiently exist

during vertebrate embryonic development. Neural crest cells give rise to dozens of cell types and are responsible for the development of autonomic and sensory nervous systems, as well as facial and cardiac morphogenesis. The enigma of the neural crest multipotency and the mechanisms partitioning the downstream fates is still on top of the list. Here, Nikaido and coauthors attempt to address the problem of a cell fate choice in the fish neural crest lineage (looking mostly at pigment cell progenitors) using a single cell technology, some functional tests and lineage tracing. Although limited, the data are strong and validated, and I feel enthusiastic and rather support the publication of this study after addressing the most critical issues.

We thank the referee for his/her general endorsement of our study.

Below I list the major concerns:

1. It would be beneficial to obtain regular single cell transcriptomics datasets for the unbiased analysis of the neural crest populations. Clustering and trajectory inference based on few preselected genes is not reliable and can be used only to address a narrow scope of questions. At the same time, the authors performed validations of gene expression, including a co-expression analysis, which makes their observations more solid. The main figure does not show the trajectory and co-expression of opposite programs/master regulators in the same progenitor cells. The mentioned trajectory is missing from the Extended Data Figure 5. Navigating such trajectory must be super hard and unreliable given these sparse data. This is a very important point of the study and should be brought up in the main figure (Figure 2), preferably after obtaining a regular 10x or Smartseq-based dataset.

We apologise for the inadvertent omission of the pseudotime lines in the main figure (Fig. 2), which is now corrected, and is now cross-referenced in the main text in the paragraph beginning 'Due to the developmental gradient along'.

*A combination of issues, including the COVID pandemic preventing access to a planned collaborator's facility, have prevented us from obtaining our own scRNA-seq data. However, the fortuitous publication of a suite of studies, many focused on the zebrafish pigment cells at some level, have provided a wealth of data here. Consequently, we have performed a meta-analysis of the combined dataset, representing 82737 cell profiles. The results of this meta-analysis is presented in a new section "Nanostring hybridization profiling generally agrees with published RNA-Seq data but can identify genes with low expression". In this section we demonstrate that the scRNA-seq and our Nanostring approaches have complementary advantages and disadvantages, so they tend to show different things. Thus, the scRNA-seq, with its quantitative detection of thousands of genes, is ideal for documenting the differentiation process in detail, but in many cases fails to detect low level gene expression likely characteristic of lineage priming and the transcriptional indication of potency that we are proposing here. In contrast, we show that our Nanostring approach is much better suited for the latter, giving detection of very low level expression; however, it provides less reliable quantitative data, making the detailed description of differentiation less informative, and even control differentiated pigment cells show detectable expression of transcription factors for other fates (e.g. *phox2bb*). As we noted before, this may reflect a relaxation of the GRN under our experimental conditions and is anyhow ideal for allowing detection of retained multipotency, but less helpful for the detailed description of differentiation which the referee discusses. It is for this reason that we have not emphasized the differentiation pathway in our study. However, we do note that the Nanostring data does show rise and loss of *tfec* and *ltk* expression in a subset of differentiating melanocytes and the opposite changes in *mitfa* transcription in the iridophores and*

melanocyte pathways. We have now expanded our summary of these issues in the Discussion in the paragraph beginning 'We reconcile our new view of NCCs retaining high multipotency'.

2. The authors write: "This provides a strong indication of the highly dynamic nature of the fate specification and differentiation process – that apparently specified cells identified *in vivo* are held in states biased towards one or more fates by environmental signals."

I believe the authors do not sufficiently discuss the essential progress made by other teams in this direction. The states of biased NCCs were carefully observed and experimentally tested in mice by Soldatov et al. 2019, which the authors cite to discuss another set of arguments. The combination of opposite genetic programs in the same individual neural crest cells was also observed by these authors. The results reported in this manuscript go perfectly well along these discoveries in mouse neural crest, and the authors shall discuss this convergence in data and ideas (co-activation and competition of programs in same cells), including the evolutionary aspect (conservation of this type of fate conflicts in fish and amniote neural crest).

We have expanded our discussion to make explicit reference to the Soldatov study of mouse neural crest (originally excluded here simply because we were focused on pigment cell development which Soldatov et al. does not address), but also to include all the other zebrafish studies published since our submission (Tataraki et al, 2021, etc), although this is necessarily brief due to the space constraints. However, as discussed in the previous point, we believe that all these scRNA-seq studies differ from ours in that they are not detecting the more weakly-expressed genes efficiently, if at all (most of the fish studies detect <3000 genes per cell; the two exemplary mouse studies detect c. 7000 genes per cell). Thus, some of the key genes (e.g. *ltk*), known to be critical for the processes of pigment cell formation, are barely detected in these other studies; this is significant because we know in the well-studied pigment cell system that low level *ltk* expression is characteristic of an initial phase in premigratory neural crest cells, where it functions in fate specification from a multipotent progenitor – indeed, we provide an experimental test of this in our manuscript. Consequently, our analysis shows that the zebrafish scRNA-seq studies appear to be inadequate to assess the evidence of broader multipotency. In this sense we would argue that our study is complementary to these scRNA-seq studies, but also identifies a somewhat underemphasised limitation of the technique; we speculate that the scRNA-seq method is highly sensitive at picking out the process of differentiation from early time-points where opposing programmes compete for dominance, but perhaps less sensitive at detecting an earlier phase when all or many programmes are in competition, but represented only by low level expression of limited number of key genes, e.g. transcription factors (note that at these stages it is far from clear that this low level expression necessarily translates to functional protein levels; but it does indicate the potential for their expression, and hence a minimum measure of cell potency). Furthermore, whilst Soldatov et al. nicely demonstrates how differentiating cells co-express two competing programmes, our data is indicating co-expression of (parts of) multiple genetic programmes, which we consider an example of lineage priming. This co-expression is characterised by variable combinations of levels of key genes, as well as by different combinations of key genes. It may be that some of this variation comes from a fundamentally dynamic process that is here captured as a series of snapshots. In our meta-analysis of the scRNA-seq data available from pigment cell studies, we note a good concordance between markers for some clusters in multiple studies. However, the low level expression of the markers used in our Nanostring study makes it difficult to correlate our cell-types with those in these other studies, where they are poorly detected outside of cells differentiating into types other than those for which these markers are characteristic in their differentiated state. Nevertheless, the zebrafish scRNA-seq data also shows co-expression of markers for more than two distinct fates in

some of these cell-types. We have now briefly summarised all this in the paragraph beginning 'We reconcile our new view of....' And in the following paragraph, beginning 'Our data reinforce certain key observations...' in which we make explicit that there is a strong convergence in our views with those from the mouse studies by the Adameyko group.

In addition, we have also added reference to Kastri et al., 2022 in EMBO J (published since we submitted our manuscript), which concludes that in mammals the Schwann Cell Progenitor state is a hub-like state that is a progenitor for multiple neural crest-derived fates; the authors tend to interpret this hub-like state as a heterogenous mixture of differentially-biased progenitors following streams of differentiation, in which cells initiate a bias state and then commit. Kastri et al. suggest resolution of pairs of alternative trajectories by preliminary activation of competing transcription programs, much as published in their previous work (Soldatov et al., 2019). It is an open question if the data could equally be reinterpreted in terms of a CFR model in which these hub cells can cycle between the biased states; this would require allowing re-entry into the same state several times along the pathway, an option which is to our knowledge poorly supported by the pseudotime ordering software.

Whilst our data are certainly consistent in many ways with that from the mammalian studies, it is clear that the expression of *ltk* in zebrafish is not conserved in mouse. However, this is consistent with its clear role in the specification and commitment of iridophores; this cell-type is not found in mammals, so the absence of expression in the neural crest is consistent with, and may even causally explain, this evolutionary change in potency. We now add brief mention of this in the Discussion in the paragraph beginning 'Our data reinforce certain key observations...'

3. Next, the authors should be careful with their conclusions as they do not test the actual fate restrictions, and instead they observe the transcriptional landscape and possible transcriptional conflicts. Although the cells show the co-expression of opposite transcriptional programs and individual master genes, they might be already fate restricted in one specific direction depending on the level of program competition, the epigenetic state, the signaling landscape and many other factors. The authors need to separate the genealogical portrait (actual situation) from the transcriptional portrait, which is merely "instrumental". The genealogical portrait can be obtained only experimentally via different types of clonal lineage tracing and perturbation experiments, and the concept of progressive or any other type of fate restrictions can be applied only to a genealogical portrait of the system. The transcriptional portrait, however, is useful to predict and understand the mechanisms providing the genealogical outcomes. This theory of genealogical vs descriptive transcriptional analysis must be extensively discussed throughout the manuscript, and the authors should interpret their data in a context of this accurate theory. This is exactly why I welcome clonal tracing and more of the functional experiments to test the cyclic fate restrictions model proposed by the authors. Without extensive functional experiments and quantitative clonal analysis of fate restrictions, the concept of CFR does not bear any significant weight beyond being a hypothesis (which is not so different from PFR concept in such unsupported case). I would really love to see the clonal data, with probability distributions, as they would make the author's point ultimately strong (some inspiration can be obtained from the reference number 56: Singh et al., Dev Cell 2016: [https://www.cell.com/developmental-cell/pdf/S1534-5807\(16\)30423-3.pdf](https://www.cell.com/developmental-cell/pdf/S1534-5807(16)30423-3.pdf)).

We respectfully disagree that the CFR model is not so different from the PFR concept; at its core the PFR model refers to stable restriction of fate potential, very different to CFR. In fact, we would argue that the CFR model is more similar to the DFR model, where cells retain full multipotency during and after migration! Of course, the model is new and requires careful testing, but these tests are not trivial and will require detailed and demanding further work. As noted above, we think our model

provides an alternative way of interpreting some of the most recent findings in the field, most notably from the key work of the Adameyko lab (e.g. Soldatov et al. and now Kastri et al., 2022). Consequently, we consider our model has value in stimulating discussion, as indeed it already has (e.g. Erickson et al., 2022, Seminars in Cell and Dev. Biol.).

A holistic view of neural crest fate specification and commitment, as discussed by the reviewer, will require integration of all the aspects the reviewer outlines, but our study highlights the possibility that results from current single cell methods, impressive as they are, likely do not routinely achieve the sensitivity needed to capture the complete picture. The Singh et al study is an important one, but it focuses on post-metamorphic pigment cells and their origin from neural crest-derived adult progenitors; indeed, it provides convincing evidence of the retained broad multipotency (neural and pigment cell) of some, and perhaps all, adult pigment cell progenitors (an observation that is readily explained within the CFR model, as we outline in our recent reviews (Kelsh et al., 2021; Dawes and Kelsg, 2021)). However, we think the approach highlighted by the referee is unlikely to be illuminating for the process of embryonic pigment cell development investigated here. Longstanding studies (including one by us) showed that, in the zebrafish, clone sizes of individual neural crest cells that adopt specific embryonic fates, including the pigment cell-types, remain very small. Consequently, we think the genealogical link the referee seeks is very difficult to demonstrate, and, at the very least, it is far from clear that fate commitment and cell division is tightly coupled in zebrafish. Indeed, a clonal study of 24 hpf *mitfa*-expressing neural crest cells concluded that post-mitotic pigment cell precursors remained plastic, becoming at least melanocytes or iridophores (Curran et al., 1994).

The model of CFR must be at least reconciled with the existing clonal genealogical data for a zebrafish pigment cell lineage reported by Singh et al., Dev Cell 2016. The transcriptional portraits shall be interpreted in this context, and possible explanations need to be provided for any identified inconsistencies.

As noted above, the specific biology of the zebrafish means that the genealogical approach utilised in the Singh et al study is not informative in the embryonic pigment cell context. Nevertheless, the Singh et al. study does indicate that neural crest cells associated with the peripheral nervous system, and likely therefore Schwann cell precursors, retain broad multipotency into adulthood, consistent with previous work that showed an association of adult pigment progenitors with the peripheral nervous system (Dooley et al., 2013; Budi et al., 2011).

4. The authors performed fate-mapping experiments from Ltk-expressing cells and found multiple progeny beyond pigment cell lineage. This suggests that the expression of Ltk is not restricting the potential of a cell lineage. At the same time, the expression of Ltk can still be a passive or active biasing factor, and this should be addressed. I wonder if the early progenitor cells expressing Ltk end up in specific pigment cell sublineage more often as compared to the cells not expressing Ltk at the early stage (in clonal analysis). The authors should work with probabilities of different outcomes depending on the early expression of a given biasing factor.

Our data using the NPM-Ltk (constitutively activated fusion protein) directly addresses the question the reviewer raises. Ltk is a receptor tyrosine kinase and we showed in the original Fig. 3 that constitutive activity of the kinase is inconsistent with melanocyte development. We have now explored this further, directly testing the implication (supported by the *ltk* loss-of-function phenotype in which iridophore fate specification fails) that activated Ltk *signalling* drives iridophore

development. Our data now shows that it does, indeed, drive iridophore development. Thus, Ltk *signalling* is an active biasing factor driving iridophore development. However, we also note that our data suggests that *ltk* expression is dynamic, not constant, since in wild-types levels are variable and only in a subset of neural crest cells, whereas in *sox10* mutants more premigratory neural crest cells express *ltk* (see Lopes et al., 2008); this idea is reinforced by the new data we now provide on the quantitation of *ltk* mRNA expression in premigratory NCCs, showing the striking variation in expression levels between cells (Extended Data Fig. 13). These deduced dynamic changes in *ltk* expression indicates (if protein levels broadly follow mRNA levels), of course, that many cells will exhibit phases when they are insensitive to the Ltk ligand, and hence will not experience Ltk *signalling*. This data is key in developing the CFR hypothesis (cycling expression of *Ltk* in HMPs changes their sensitivity to Ltk ligands, so that they become biased towards iridophore fate (Ltk+) or not (Ltk-); where Ltk signaling is prolonged, this drives adoption of the iridophore fate), so we have expanded our explanation in the paragraphs beginning 'We have been unable to identify clusters that' and 'To reconcile our new observations with those' to clarify this.

5. Again, a systematic clonal analysis (fate distribution in single clones traced from a biasing factor / master regulator) will be highly instrumental.

The reviewer suggests an experiment that has been revealing in other systems (e.g. analysis of *neurog1*-expressing cells in mouse; Soldatov et al., 2019). Unfortunately, previous work from our and others' labs shows that the small clone size of zebrafish neural crest cells makes the results difficult to interpret. Thus, by c 24 hpf essentially all neural crest cells express *mitfa* (Curran et al., 2009, Dev.Biol.), and yet labelling cells between 18 and 24 hpf gives rise to a very large majority of clones consisting of a single pigment cell fate (Raible and Eisen, 1994; Dutton et al., 2001). However, the clone size is so small (1-4 cells; Raible and Eisen, 1994) that it is impossible to tell whether this results from the labelled cells being committed already or simply that the clone size is so small that differential exposure of progeny to different environmental signals becomes limiting. Furthermore, these clonal labelling studies at 24 hpf show that these *mitfa*-expressing cells generate both melanocytes and iridophores (Curran et al., 2010).

6. Can the authors use the power of a zebrafish system to bias or re-bias NCCs in the context of competing programs?

We have done exactly this. Thus, we had originally shown that the constitutively-activated Ltk fusion protein (NPM-Ltk) strongly biases cells away from a melanocyte fate (Fig. 3). Subsequently, we have now shown that expression of the constitutively-active Ltk protein drives cells to adopt an iridophore fate at the expense of other pigment and neural fates, consistent with our suggestion that Ltk signalling drives iridophore fate specification (and subsequent differentiation) (new Extended Data Fig. 15).

Again, I feel enthusiastic about this story and hope to see it again after all improvements.
Thank you!

Reviewer #3 (Remarks to the Author):

Here, Nikaido and Subkhankulova, et al uncover a cyclical fate restriction mechanism that describes the development of zebrafish pigment cells from a multipotent neural crest intermediate state into

differentiated cell states. Using single cell Nanostring profiling, they were unable to recover a partially restricted intermediate population suggesting fate restriction arises from a highly multipotent intermediate population. Further, the authors investigate the biphasic expression of *ltk*, which marks multipotent neural crest as well as differentiated iridophores. By inhibiting or activating *Ltk*, the authors test the multipotency of *ltk*-expressing cells. This manuscript presents a new model for fate specification of pigment cells, which should be investigated in other neural crest cell types in the future. The following concerns should be addressed prior to publication:

Main:

1. With such a limited gene set for Nanostring (45 genes), the scope of understanding the fate restriction mechanism is limited to the genes we already know. How were the 45 genes chosen for the probe set?

As deduced by the reviewer, the 45 genes were chosen to be most informative regarding the fate restriction mechanism. Firstly, they include all genes known at the time, based on mutant phenotypes, to play crucial, cell-type specific roles in fate specification of each of pigment, neuronal and glial cell fates. They also include selected differentiation markers identified from the literature, but also from the bulk sequencing study by Higdon et al.. Given the specific focus here on the Progressive Fate Restriction model of pigment cell development, we ensured that melanocyte and iridophore genes were especially well-represented, to enhance sensitivity to the chromatoblast and melanoiridoblast intermediates that had been predicted in the literature. Finally, they include selected markers for the very earliest stages of neural crest development. We also included a housekeeping gene for normalisation. Note that *gapdh* (a second housekeeping gene) was included in the probe sets, but the data showed it to be expressed at lower levels and rather more variably than expected, so that we preferred *rlp13* for normalisation. The key information are briefly noted in the main text (First paragraph of section 'Single cell transcriptional profiling of zebrafish NCCs....') and compared in detail in Supplementary Table 1. We have now added a statement to the Supplementary Table 1 covering these comments.

2. (Line 261) The number of iridophores should be quantified as well. Were there differences in the number of neurons after altered *Ltk*?

We did not score for iridophore fates in this assay because these cells are intrinsically autofluorescent, making identification of cells expressing (GFP+) the activated or control (DK) constructs troublesome. However, we now provide data that cells expressing the activated *Ltk* signaling construct drives the expression of *tfec* at the expense of key fate-specific transcription factors for other pigment and neural fates, consistent with a quantitative shift to iridophore fate (new Extended Data Fig. 15).

3. How does constitutive LTK affect neurons and glia?

The reviewer raises an interesting question, whether or not LTK signalling is compatible with neural fates? To address this, we expanded our study of the impact of constitutively-activated *Ltk* signaling within the NCCs, to examine the impact on *phox2bb* expression, since *Phox2bb* plays a crucial role in autonomic neuron fate determination. As we show in a new Extended Data Fig. 15, activated *Ltk*

signalling represses *phox2bb* expression. Note that we also assessed *neurog1* expression, but this is detected in too small a sample to assess whether or not it might also be repressed.

4. Can the authors comment on how Schwann Cell Precursors may confound their results and cyclical model as they are known to remain multipotent later into development.

We do not see Schwann Cell Precursors as confounding our results, nor our model. Indeed, as noted in the Discussion (see paragraph beginning 'We have been unable to identify clusters....'), we see them as (at least part of) the explanation for why we see Highly Multipotent Progenitors (HMPs) at unexpectedly late stages, e.g. 72 hpf): It is the identification of SCPs as retaining multipotency that explains the result and helps validate our data. This is why we were both particularly surprised, but also reassured, to see occasional *phox2bb* expression in just such cells (Fig. 2f-h). As we have discussed elsewhere (Kelsh et al., 2021, Development), we see environmental influences on neural crest cells as critical in imposing various transcriptional states upon the HMPs. One such state is the SCP. This can be considered a differentiated cell state (albeit perhaps not a fully differentiated one), with a phenotype driven at least in part by neuronal signals received from the adjacent axon. Under our Cyclical Fate Restriction model we propose that these cells will not be actively cycling but, importantly, we predict that on release from the axonal differentiation signals, they will re-enter a cycling state, consistent with their activation as multipotent stem cells, and with a cycling behaviour influenced by local signals to bias their fate options appropriately.

5. Fig2: More stages should be investigated with FISH

In line with the reviewer's suggestion we have expanded our analysis of *ltk* and *phox2bb* coexpression beyond the 27 hpf stage shown in Fig. 2, to now include each of the following stages: 24, 30, 36 and 50 hpf and 3 dpf. These data are now shown in Extended Data Figure 11 and discussed in the section 'Co-expression of key fate specification genes in situ'.

Minor:

1. FAC-sorted, not FACS sorted

We have replaced 'FACS sorted' with 'cells isolated by FACS'

2. The authors mention 8 time points were isolated (Line 84), what are these time points? Only 7 are shown in Fig2.

The time-points were 18, 21, 24, 30, 36, 48, 60 and 72 hpf. So the original main text was correct. Fig. 2 does not actually show any time-points, but simply identifies 7 clusters of cells defined by their expression profiles.

REVIEWERS' COMMENTS

Reviewer #1 (Remarks to the Author):

The authors have carefully addressed the concerns of the previous review. They added significant additional data to support their ideas. Because of this, the paper has become more dense, so the authors should go through the writing to be sure it is clear. In addition, I am still not completely convinced of the cylindrical fate model and how *Ltk*, a gene required for pigment cell fate, fits. However it is worth putting the model out to the community for continued discussion. A few minor comments:

1. Fig 3 panels are not updated, please check the inset labels
2. In Extended data figure 12, please make clear what the *gfp* is here
3. Some of the new figures are extremely small ie Extended data figure 6
4. There is not much discussion about the *sox10* mutant nacre sequencing. It is curious that in the nacre mutant, neural crest are stuck in an earlier state. Given the impact of this, I would encourage a bit more of a discussion on this point.

All this said I encourage publication. This paper is significantly better, presents novel data and I feel it is worth publishing and putting it out into the community for continued discussion.

Reviewer #2 (Remarks to the Author):

The authors responded to all my comments with sufficient rigor. I am happy to proceed with publication.

Response to Reviewers NCOMMS-21-24407B

We are grateful to all the reviewers for their helpful queries and suggestions. We have addressed the final ones of Reviewers 1 and 2 as documented here. All changes from the previous version are highlighted in red in the revised manuscript submitted.

REVIEWERS' COMMENTS

Reviewer #1 (Remarks to the Author):

The authors have carefully addressed the concerns of the previous review. They added significant additional data to support their ideas. Because of this, the paper has become more dense, so the authors should go through the writing to be sure it is clear. In addition, I am still not completely convinced of the cylindrical fate model and how *Ltk*, a gene required for pigment cell fate, fits. However it is worth putting the model out to the community for continued discussion. A few minor comments:

We have taken the opportunity to carefully examine the manuscript and revised where we thought we could increase clarity, whilst conforming to the word count requirements for *Nature Communications*. In particular, we have sought to clarify the relationship between *Ltk* and the Cyclical Fate Restriction model.

1. Fig 3 panels are not updated, please check the inset labels

Apologies. We had revised the original figure, but had not replaced the small version in the submitted Tables/Methods/Fig legends with figures document examined by the reviewer. This has now been corrected.

2. In Extended data figure 12, please make clear what the gfp is here

We have now clarified that this is enhanced GFP.

3. Some of the new figures are extremely small ie Extended data figure 6

We believe the reviewer is referencing Ext Data Fig 7, which is very small in the submitted document, and have now revised this; we have also taken the opportunity to modify Extended Data Fig 8 in an equivalent manner. We have changed gene labels to much larger font and moved them to the bottom right of each graph, so that they are more legible. We have ensured that the resolution of the submitted Supplementary Information is sufficient that these figures are fully legible when expanded on screen.

4. There is not much discussion about the *sox10* mutant nacre sequencing. It is curious that in the nacre mutant, neural crest are stuck in an earlier state. Given the impact of this, I would encourage a bit more of a discussion on this point.

We have expanded discussion of the *sox10* mutant (NB this is not *nacre*, which is the *mitfa* gene), to emphasise the consistency with the interpretation made in our earlier studies. We have been

reluctant to speculate too much at this stage, due to the limitations in terms of marker numbers, instead preferring to highlight a need for further characterisation of the mutant cells.

All this said I encourage publication. This paper is significantly better, presents novel data and I feel it is worth publishing and putting it out into the community for continued discussion.

Thank you.

Reviewer #2 (Remarks to the Author):

The authors responded to all my comments with sufficient rigor. I am happy to proceed with publication.

Thank you.